# A pair of atypical NLR-encoding genes confers Asian soybean rust resistance in soybean

Qingnan Hao [1,2,6], Hongli Yang [2,6], Shuilian Chen [3,6], Chanjuan Zhang [1], Limiao Chen[1], Dong Cao [1], Songli Yuan [2], Wei Guo [2], Zhonglu Yang[3], Yi Huang[3], Yanhui Qu [4], Lucy Qin[5], Xiaoyan Sheng [5], Xueyan Wang[5], Chandrani Mitra [5], Heng Zhong [5], John Dawson [5], Eric Bumann [5], Wenling Wang[5], Yaping Jiang[5], Guozhu Tang[5], Ryan Carlin[5], Haifeng Chen [3] ✉, Qingli Liu [5] ✉, Zhihui Shan [2] ✉ & Xinan Zhou [1] ✉

Asian soybean rust (ASR), caused by *Phakopsora pachyrhizi*, is a devastating disease that is present in all major soybean-producing regions. The limited availability of resistant germplasm has resulted in a scarcity of commercial soybean cultivars that are resistant to the disease. To date, only the Chinese soybean landrace SX6907 has demonstrated an immune response to ASR. In this study, we present the isolation and characterization of *Rpp6907-7* and *Rpp6907-4*, a gene pair that confer broad-spectrum resistance to ASR. *Rpp6907-7* and *Rpp6907-4* encode atypic nucleotide-binding leucine-rich repeat (NLR) proteins that are found to be required for NLR-mediated immunity. Genetic analysis shows that only Rpp6907-7 confers resistance, while Rpp6907-4 regulates Rpp6907-7 signaling activity by acting as a repressor in the absence of recognized effectors. Our work highlights the potential value of using *Rpp6907* in developing resistant soybean cultivars.

Soybean, a crop of great significance, serves as a source of oil and protein for both human and animal consumption on a global scale (http://faostat.fao.org/). Asian soybean rust is a highly destructive disease that ranks among the top five biotic threats to agriculture[1,2]. The impact of soybean rust on yield can be as high as 40–80%, with even a low incidence rate of 0.05% posing a significant risk to farmers and agricultural systems. The disease is ubiquitous in the soybean growing areas of Latin America, where 210 million metric tons of soybean are projected to be produced in 2022/23 (https://apps.fas.usda.gov/psdonline/app/index.html)[3], and on average representing a gross production value of U.S. $115 billion per season (https://www.ers. usda.gov/data products/season-average-price-forecasts.aspx)[3]. Chemical control is the only method by which to control the disease due to the limited resistant soybean germplasm. In Brazil alone, the cost is estimated to be approximately U.S. $2.8 billion per harvest for managing *Phakopsora pachyrhizi* (https://news.agropages.com/News/NewsDetail--45811.htm)[3]. The recent number of cases of ASR in the 2022/23 harvest exceeded 119 in nine Brazilian states, and this was an increase of more than 220% in just one season compared to the same period last year between November 2022 and January 2023 (https://news.agropages.com/News/NewsDetail--45811.htm)[2]. In addition to lowering yields, plant diseases also result in loss of species diversity,

[1]Soybean Genetics and breeding team, Institute of Oil Crops Research, Chinese Academy of Agriculture Sciences, Wuhan 430062, China. [2]Chinese Academy of Agricultural Sciences/Key Laboratory for Biological Sciences of Oil Crops, Ministry of Agriculture and Rural Affairs, Wuhan 430062, China. [3]Soybean Collaborative Innovation Center, Institute of Oil Crops Research, Chinese Academy of Agriculture Sciences, Wuhan 430062, China. [4]The Graduate School of Chinese Academy of Agricultural Sciences, 100081 Beijing, China. [5]Syngenta Crop Protection, LLC, Research Triangle Park, Durham, NC 27709, USA. [6]These authors contributed equally: Qingnan Hao, Hongli Yang, Shuilian Chen. ✉e-mail: chenhaifeng@caas.cn; qingli.liu@syngenta.com; shanzhihui@caas.cn; zhouxinan@caas.cn

mitigation costs due to control measures, and downstream impacts on human heath[4].

To date, eight major dominant resistance loci (i.e., Rpp1[5], Rpp2[6], Rpp3[7], Rpp4[8], Rpp5[9], Rpp6[10], Rpp6907[11], and Rpp7[12]), have been identified and mapped on the soybean genome, but none of the genes have been cloned. Only CcRpp1 from pigeon pea genome has been cloned[13]. Whereas, the utilization of CcRpp1 for enhancing soybean genetics remains a distant prospect due to its exclusive deployment through transgenic methods, which are subject to extensive regulatory procedures. Additionally, the emergence of resistance-breaking ASR field populations has rendered cultivars with major resistance loci from soybean increasingly ineffective[14]. This lack of cloned resistant genes hinders the comprehension of their molecular foundation and their use in soybean breeding through molecular marker-assisted selection and gene engineer. In order to achieve durable and broad-spectrum resistance to ASR, there is an urgent need to identify new resistant genes[15].

Novel R genes with enhanced efficacy against predominant pathogen races can potentially be discovered in wild species and landraces[16]. Among these, the Chinese soybean landrace SX6907 has been identified as possessing complete resistance against Phakopsora pachyrhizi SS4, surpassing other resistant germplasms[11]. The Rpp6907 locus, previously mapped to a 110.9 kb region on Chr18, is different from Rpp1 in PI 200492, PI 561356, PI 587880A, PI 587886, and PI 594538A, or allele of Rpp1-b[11].

In this research, we clone the Phakopsora pachyrhizi resistance gene Rpp6907-7 in SX6907. Additionally, we identify its corresponding sensor NLR Rpp6907-4. Our findings indicate that Rpp6907-7 and Rpp6907-4 are essential components of NLR-mediated immunity and represent an atypical NLR pair. Through genetic analysis, we determine that the resistance conferred by Rpp6907-7 is independent of Rpp6907-4, which instead functions as a negative regulator of Rpp6907-7 signaling in the absence of recognized effectors.

## Results

### Rpp6907-7 is the R gene for ASR

To clone Rpp6907, three simple sequence repeat (SSR) markers, SSR24, SSR40, and SSR32[11] were used to screen the SX6907 bacterial artificial chromosome (BAC) library. Subsequently, two positive BAC clones were identified and assembled into a 174.9 kb contig that encompassed the Rpp6907 locus (Fig. 1a). The Rpp6907 region was annotated and compared between the SX6907 and Williams 82 genome assemblies, revealing a larger interval of 150.22 kb in SX6907 as compared to the 93.45 kb interval in Williams 82. In summary, a 56.77 kb insertion was observed in SX6907, which contained additional genes and repetitive elements not present in the Williams 82 genome assembly (Fig. 1a). The ~60 kb Rpp6907 region from SX6907 is situated between the genes Glyma.18G283100 and Glyma.18G283300 in the SX6907 genome. Notably, R1 to R6 NLR encoding genes are aligned in close proximity, forming a cluster, while a partial duplication of the Glyma.18G283100 gene is found preceding the R7 (NLR) gene (Fig. 1a). Southern blot analyses, using the P-loop region present in the seven NLR genes (R1, R2, R3, R4, R5, R6, and R7) as a probe, confirmed that the Rpp6907 locus in SX6907 contains seven NLR genes (Supplementary Fig. 1a). A phylogenetic tree of the complete NLR genes showed that R7 and R1 are the most closely related members of this cluster (Supplementary Fig. 1b, c). Of these, R1, R2, R3, R5, R6, and R7 are predicted to encode NLR proteins containing 773–977 amino acids; R4 is truncated and predicted to encode an NLR containing 534 amino acids with an incomplete CC domain and shorter LRR domain. We cloned and sequenced these candidate genes in a soybean collection including Tianlong 1, Williams82, wild soybean, landraces, and breeding cultivars. The findings indicate that solely R7 exhibited specificity towards SX6907, while the remaining six genes were capable of being cloned in certain susceptible soybean cultivars. These results imply

that R7 may be the crucial gene in conferring resistance to the rust pathogen. A multiple sequence alignment was performed on the gene with the highest sequence similarity to R7 in 13 soybean varieties to examine the mutations present. The analysis revealed that R7 exhibited variations in two brief regions, including a 15-bp insertion, a 9-bp deletion, a 6-bp deletion, a 3-bp deletion, and 36 single nucleotide polymorphisms (SNPs) (Supplementary Fig. 2; Supplementary Data 1).

In order to ascertain the potential resistance conferred by R7 against Phakopsora pachyrhizi, we produced transgenic Tianlong1 plants that expressed R7 under its native promoter. The resulting transgenic lines, namely Rpp6907.2, Rpp6907.3, and Rpp6907.6, exhibited expression of R7 (Supplementary Fig. 3a) and demonstrated resistance to ASR isolate SS4. The subsequent generation showed that the resistance phenotype co-segregated with the transgene R7 (Fig. 1b). The R7 lines, both heterozygous and homozygous, exhibited notable resistance to ASR. Upon analysis of the homozygous plants, the presence of the transgene was found to be significantly associated with a reduction of over 99% in lesion counts per unit leaf area. Hemizygous plants, on the other hand, displayed reddish-brown (RB) lesions without any sporulations 14 days post infection and demonstrated a reduction of 60 - 70% in lesion count per cm². Null plants, however, exhibited TAN-type lesions, which were tan-colored and had a similar level of sporulation as susceptible controls 14 days after inoculation[7,17,18], which is typical of a susceptible reaction to Phakopsora pachyrhizi (Fig. 1b; Supplementary Data 2). An R7 RNAi plasmid was constructed, and three independent transgenic events were obtained by introducing the plasmid into the resistant cultivar SX6907, exhibited expression of R7 (Supplementary Fig. 3b) and demonstrated resistance to ASR isolate SS4. The resistance level was determined in R7 RNAi transgenic plants (Line 4, Line 5 and Line 7), and the plants exhibited a susceptible phenotype (Fig. 1b). The outcomes of both over-expression and RNAi transgenic experiments consistently verified that the NLR encoding gene R7 corresponds to the Rpp6907 gene, which was consequently renamed Rpp6907-7.

In order to assess the spectrum of ASR resistance of Rpp6907, two additional transgenic lines were produced to express Rpp6907-7, utilizing a distinct susceptible line, 06KG, which is a proprietary line of Syngenta. Homozygous $T_2$ or $T_3$ plants were evaluated from three $T_0$ events, with 14 field populations from the U.S. and Brazil. The transgenic lines harboring Rpp6907-7 demonstrated nearly complete immunity to all ASR populations that were tested (Fig. 1c, Supplementary Data 3), confirming that Rpp6907-7 alone is responsible for the broad spectrum of resistance carried by SX6907. Tianlong1 and 06KG exhibit distinct genetic backgrounds, yet Rpp6907-7 confers comparable levels of resistance via the endogenous cassettes in both backgrounds, indicating minimal host dependency of Rpp6907-7. Furthermore, the Rpp6907-7 events exhibit no discernible negative phenotype in a potted cultivation test (Fig. 1d, e). No significant difference in plant/row and plot were observed between Rpp6907-7 containing plants and null/wild type plants when visually inspected during the growing season, or when their yields were compared to non-genetically modified plants (Fig. 1f, g). Therefore, the utilization of Rpp6907-7 holds significant potential in the fields of breeding and genetic engineering for ASR resistance, as it effectively preserves the equilibrium between disease resistance and yield.

### Evolutionary history of the NLR encoding genes at the Rpp6907 locus

In order to comprehend the diversification of the NLR encoding gene cluster within a range of soybean genomes, a phylogenetic tree was constructed utilizing data from the Rpp6907 region of 28 soybean accessions' high-quality genome assemblies[19–21]. The Rpp6907 region was segregated into three major haplotype groups (Fig. 2a). Following this, a total of 133 NLR encoding genes in the genomic regions

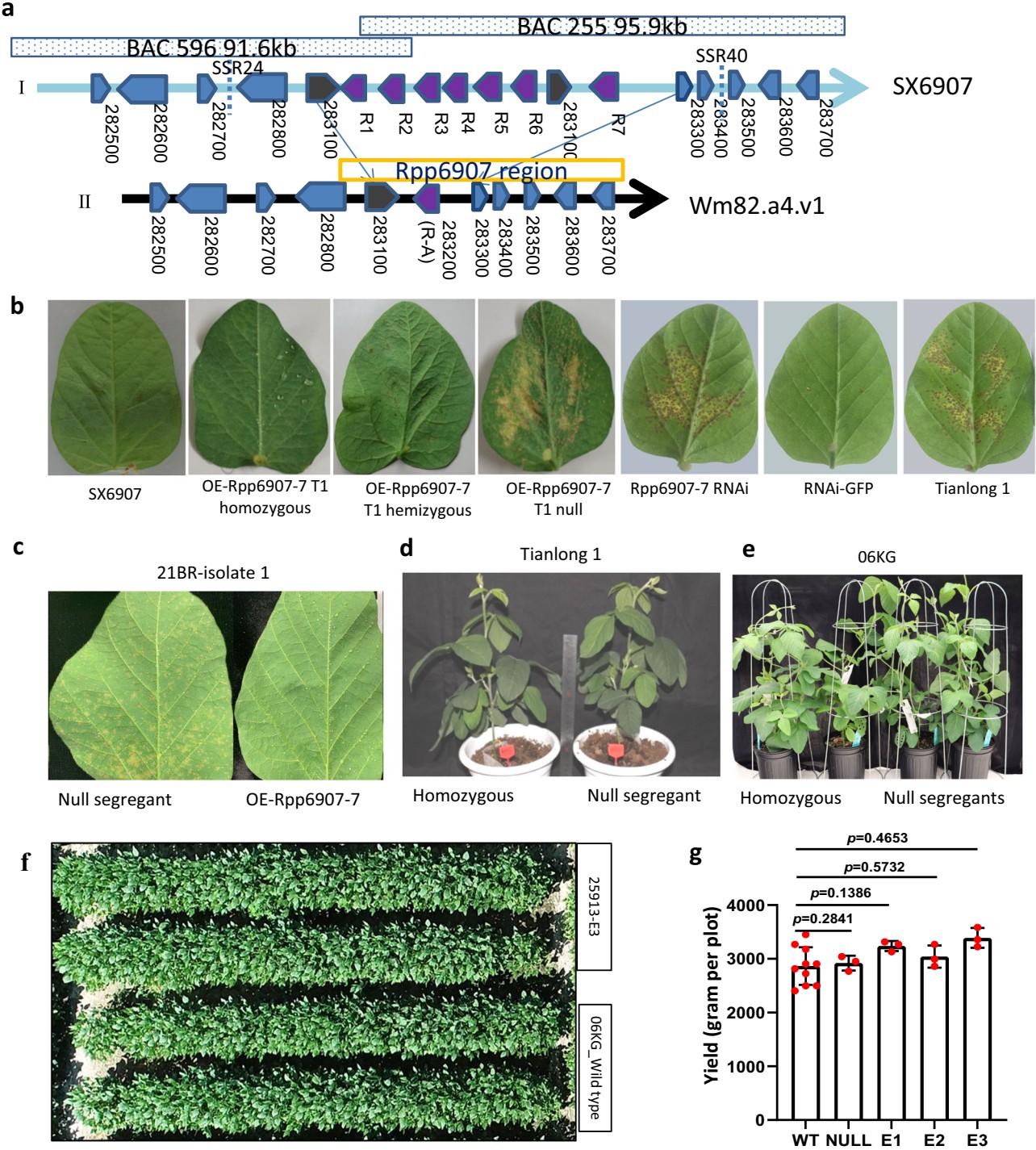

**Fig. 1 | Cloning and functional characterization of Rpp6907-7. a** Genetic and physical maps of *Rpp6907*. (I) Graphical representation of the SX6907 annotated sequences. (II) Comparison of orthologous regions in the reference genome Wm82.a2. v1. **b** Pathogen infection assay using SX6907, $T_1$ homozygous Tianlong1, $T_1$ Hemizygous, $T_1$ null, $T_1$ Rpp6907-7-RNAi transgenic SX6907 plant, RNAi-GFP, and Tianlong 1. The representative images from a single replicate of ten independent experiments are shown. **c** *Rpp6907-7* $T_2$ events confer complete spectrum resistance to all ASR populations (U.S. and Brazil strains) tested under greenhouse conditions. The representative images from six independent experiments are shown. **d, e** The *Rpp6907-7* events showed no obvious negative phenotypes in two

genetic backgrounds. The representative images from a single replicate of three independent experiments are shown. **f** *Rpp6907-7* $T_2$ events showed no obvious negative phenotypes in field conditions. One plot with two rows per expression cassette and null or wild type were photographed at 5 m above the plant canopy in the R2-R3 stages. Each genetic background was conducted with four replicates. **g** *Rpp6907-7* $T_2$ (E1–E3) events showed no significant difference in yield (in grams) compared to null and wild type in field conditions in complete random design. Data are shown as means ± SD ($n = 10$ replicates in WT, $n = 3$ replicates in null, E1–E3). Statistical analysis was performed using a two-tailed Student's $t$ test. Source data are provided as a Source Data file.

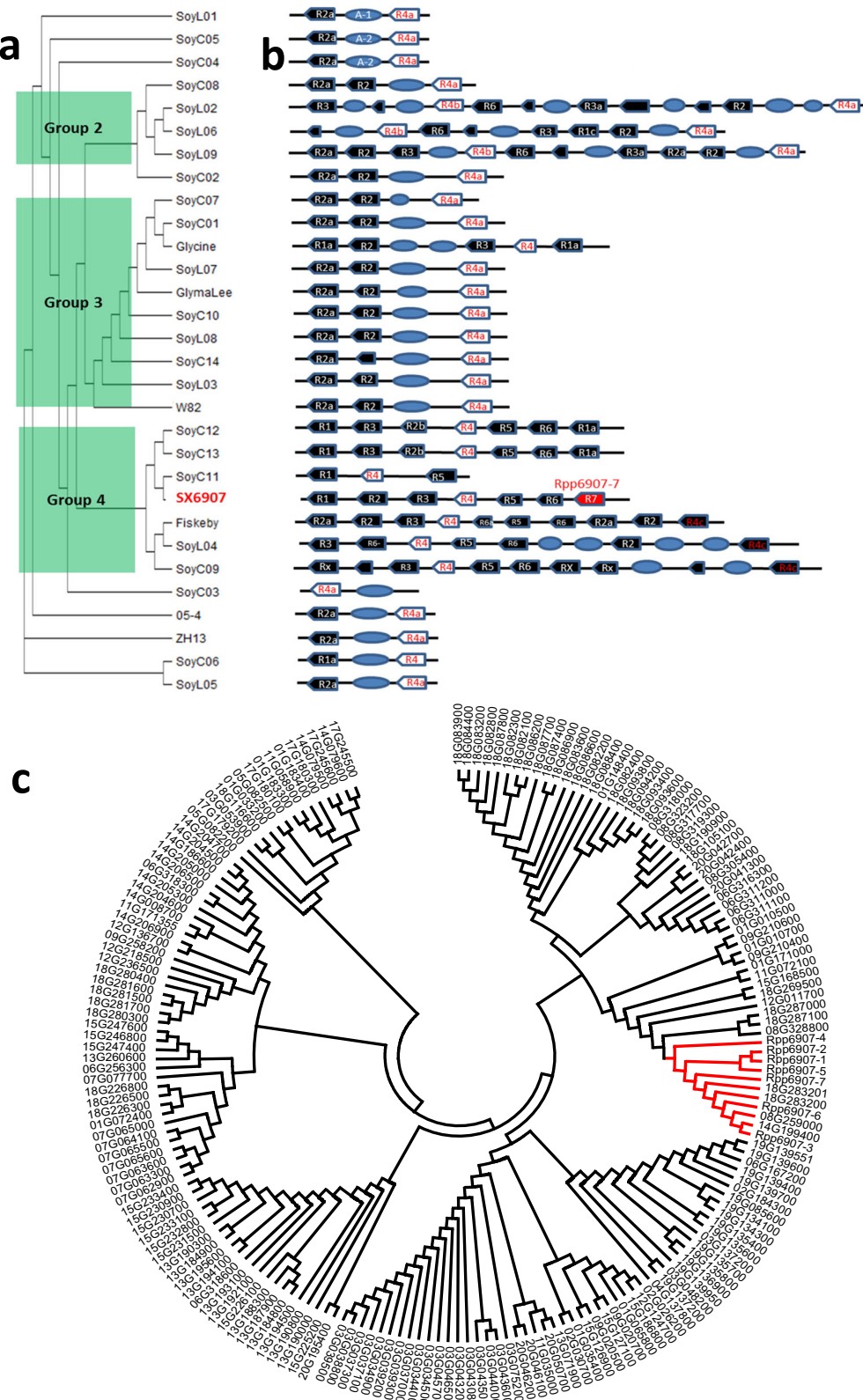

**Fig. 2 | Evolutionary of the NLR encoding genes at the *Rpp6907* locus. a** A phylogenetic tree of the *Rpp6907* locus across 30 diverse soybean genomes. Green is used to highlight the three major haplotype groups. **b** Diversification of the NLR encoding gene cluster across 30 diverse soybean genomes. Each black shape represents an NLR encoding gene. The blue ellipse represents a transposable element. **c** Phylogenetic analysis of *Rpp6907* and all known CNL genes in soybean. The clade highlighted in red spotlights the Rpp6907 branch.

corresponding to the *Rpp6907* locus were annotated. The observed copy number variation of NLR encoding genes among these varieties ranged from one to 10 copies (Fig. 2a). This finding suggests that the clustered organization and arrangements of NLR encoding genes in SX6907 played a role in generating unique resistance specificities in response to pathogen selection during the domestication process. The observed variation in NLR encoding genes might be predominantly attributed to the occurrence of illegitimate recombination, facilitated by the insertion of transposable elements (Fig. 2b). The resultant NLR proteins may exhibit enhanced compatibility with diverse pathogen recognition mechanisms and downstream defense signaling pathways, thereby contributing to their evolutionary success. A higher frequency of copy-number variants was found in NLR encoding genes compared with the whole genome. Soybean has 188 annotated CNL (coiled-coil NLRs) genes distributed across multiple chromosomes. Evolutionary analysis of *Rpp6907* and other soybean CNL resistance genes revealed that *Rpp6907* belongs to a separate clade (Fig. 2c), and this might be the result of the adaptation of soybean to new biotic and abiotic stresses to ensure a stable coevolutionary interaction between plant and environmental conditions.

### Characterization of *Rpp6907-7*

*Rpp6907-7* contained a 2571 bp exon and no introns, encoding an NLR protein of 857 amino acids (Fig. 3a). We examined basal *Rpp6907-7* expression in different tissues of SX6907 plants and found that *Rpp6907-7* was expressed in all tissues; the expression level in the young leaves was the highest (Fig. 3b). The temporal patterns of *Rpp6907-7* expression were examined both pre- and post-inoculation with SS4, revealing that the expression of Rpp6907-7 was not affected by inoculation 48 h before (Fig. 3c). The subcellular localization of Rpp6907-7 was examined using the transient expression of an Rpp6907-enhanced green fluorescent protein (EGFP) fusion construct in *N. benthamiana* leaf cells. We found that Rpp6907-7 likely localized to the plasma membrane (PM) (Fig. 3d), similar to RPM1[22], TM-2[2][23], RPS5[24] and ZAR1[25], PM localization may require for its hypersensitive response(HR) activity.

NLRs shared many characteristics when they were firstly discovered, but intensive studies have revealed a great deal of divergence in their structure and function[26]. When transiently overexpressed in *N. benthamiana*, the full length Rpp6907-7 can automatically trigger HR signaling (Fig. 3e), while none of the individual domains or fragments of Rpp6907-7 were found to induce HR upon deletion analysis of the various domains (Supplementary Fig. 4). These results suggest that Rpp6907-7, if overly expressed, may trigger a HR in a heterologous system. Site-specific mutations analysis on CC, NB-ARC, and LRR domains of Rpp6907-7 (Supplementary Data 4) and transiently overexpressed them in *N. benthamiana* has found that mutation at amino acid position 84 (Q to R) in the CC domain abolished HR by Rpp6907-7 (Fig. 3e; Supplementary Fig. 5), suggesting that the CC domain may be critical and for HR signaling. We then evaluated *Rpp6907-7* sequences similar genes in 83 Chinese soybean collections and found no orthologs of Rpp6907-7 at amino acid position 84 (Supplementary Fig. 6). Self-association and/or oligomerization of NLR proteins or their N-terminal domains has been demonstrated to be necessary for immune signaling in animals and plants[27–31]. The yeast two-hybrid and BiFC results indicated that Rpp6907-7 could self-associates through its CC domains, but not its LRR domains (Fig. 3f). However, we found Rpp6907-7 CC domains self-associates wasn't lost in 84 (Q to R) mutation (Supplementary Fig. 7).

### Rpp6907-4 represses Rpp6907-7 autoactivation in *N. benthamiana*

Plant NLR activity is regulated by intramolecular or intermolecular interactions[32]. In intramolecular model, R proteins should be maintained in an inactive state in the absence of pathogen, whereas, after

avirulence (AVR)-recognition, they are activated and induce disease resistance signaling[24,27,33]. In this study, it was observed that the transient expression of Rpp6907-7 in a heterologous system (*N. benthamiana*) without AVR resulted in the occurrence of HR. This finding suggests that Rpp6907-7 may exert its function through intermolecular regulation, specifically via NLR pairs, in soybean. It is worth noting that *N. benthamiana* lacks the presence of these regulatory factors. Ectopic expression of helper NLRs, such as RRS1/RPS4, RGA5/RGA4 and PigmS/PigmR, results in auto-activity, which can be suppressed by co-expressing the matching sensor NLRs[34–36]. In order to find the sensor of Rpp6907-7, all genes at the *Rpp6907* locus were investigated, and we found that the co-expression of Rpp6907-7 and Rpp6907-4 (R4) attenuated or completely abolished Rpp6907-triggered HR (Fig. 4a, Supplementary Fig. 8). The degree of HR inhibition was contingent upon the proportion of Rpp6907-7 and Rpp6907-4-carrying *Agrobacterium* present in the infiltration inoculum. When utilizing a ratio of 1:2, either no symptoms or only mild chlorosis were detected (Fig. 4a).We also found that Rpp6907-7-mediated resistance was no reduction in TRV:Rpp6907-4-treated SX6907 (Supplementary Fig. 9). *Rpp6907-4* contained 1603 bp exons and no introns, encoding an NLR protein of 534 amino acids (Fig. 4b, Supplementary Data 5). Rpp6907-4 was an NLR with distinct expression and subcellular localization patterns in comparison to Rpp6907-7, was discovered to have its expression stimulated by rust infection, with the highest expression observed in leaves one hour after being inoculated with Phakopsora pachyrhizi (Fig. 4c). In their resting states, sensor NLRs locate themselves to various sites in the cell, possibly to engage with appropriately localized effectors. Rpp6907-4 localizes to the cytoplasm and nucleus (Fig. 4d), co-expression of Rpp6907-7 and Rpp6907-4 shows that their subcellular localization was altered, and they could co-localize on the PM (Fig. 4e). Thus, our analysis suggests that Rpp6907-7 and Rpp6907-4 are an NLR pair, and Rpp6907-4 as the signaling NLR and Rpp6907-7 as the receptor NLR. We also found that Rpp6907-7 and Rpp6907-4 interact through the Rpp6907-7 CC domain and Rpp6907-4 C-terminal domain in a yeast two-hybrid assay (Fig. 4e). Rpp6907-7 and Rpp6907-4 could also dimerization in planta (Fig. 4f). These results show that the NLR proteins Rpp6907-7 and Rpp6907-4 interact functionally and physically to confer disease resistance. Different from the three NLR pairs that have been found[34–36], Rpp6907-4 is a truncated NLR with a short CC domain, and without additional integrated domain, such as WRKY domain in RRS1, and HMA domain in both RGA5 and Pik-1[34,37–42]. Therefore, we infer that Rpp6907-7/ Rpp6907-4 is a different kind of atypical NLR pair, which is likely to show a different recognition pattern for effectors. This also helps to enrich the complex functional principles of NLR-mediated immunity.

## Discussion

A significant breeding objective for improving crop disease resistance is the improvement of *R* genes that control ETI[43]. Due to the tradeoff between growth and defense, disease resistance, particularly when associated with high levels of *R* gene expression, can result in fitness costs[15,44]. *Rpp6907-7* was derived from a soybean landrace that has been used by local growers. Our observations have shown that transgenic soybean with homologous expression of *Rpp6907-7* confer ASR resistance while maintaining stable yield (Fig. 1g). We undertook a detailed analysis of 28 soybean accessions in the region of the *Rpp6907* locus; all varieties contained the *Rpp6907-4* or Rpp6907-4-like gene (Fig. 2b). We found that the co-expression of *Rpp6907-7* and Rpp6907-4-like gene (*W12*) from Tianlong1 also attenuated or completely abolished Rpp6907-triggered HR (Supplementary Fig. 10). We predict that *Rpp6907-4* has been retained during evolution as a protective mechanism for plant growth. Further studies are needed to decipher the full regulatory network of Rpp6907-7/Rpp6907-4-mediated resistance.

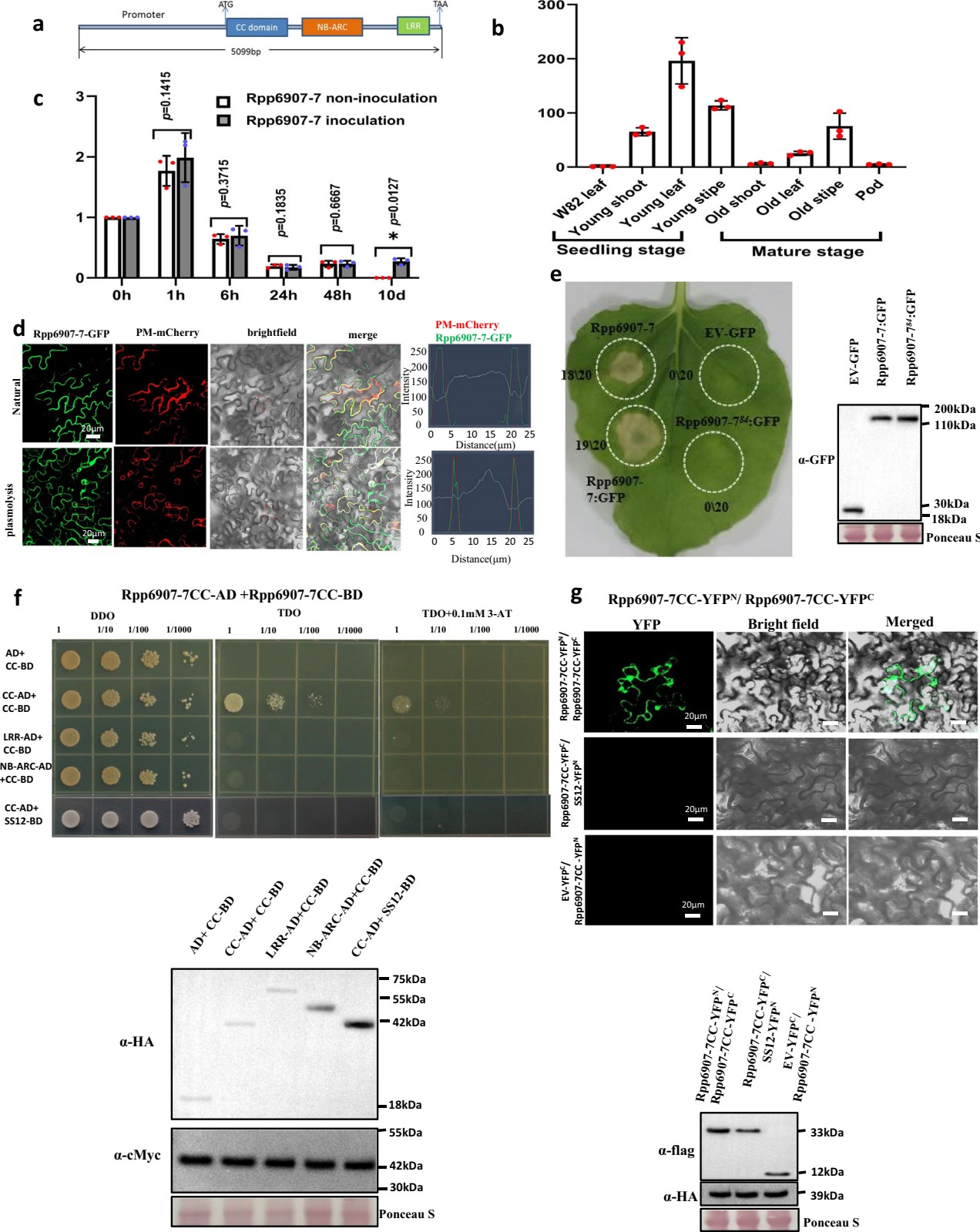

The preservation and improvement of the worldwide food supply is a significant obstacle confronting human civilization in the 21st century. The magnitude of biomass and yield loss attributed to plant diseases is substantial. ASR poses a significant hazard to soybean production due to the scarcity of disease-resistant cultivars and the destruction it inflicts. The creation of highly resistant plants to manage plant diseases is a cost-effective and efficacious approach. This article presents the initial cloning of the broad-spectrum ASR resistance gene pair *Rpp6907-7/Rpp6907-4* from soybean. Rpp6907-7 triggers an AVR-independent HR that is repressed in the presence of Rpp6907-4, which resides at the *Rpp6907* locus. They function as an atypical genomically linked pair. *Rpp6907-7* acts synergistically with *Rpp6907-4* to maintain a balance between yield and plant resistance. The exploration of the molecular mechanisms that underlie the trade-off between defense

**Fig. 3 | Molecular characteristics, tissue distribution, and protein structure of Rpp6907-7. a** Structure of Rpp6907-7 used for the complementation assay. **b** Expression of *Rpp6907-7* in different tissues of SX6907 plants. Values are the means ± SD (*n* = 3 repeats with five individuals per sample). **c** Expression of *Rpp6907-7* in SX6907 after inoculation with SS4. Values are the mean ± SD (*n* = 3 repeats); *P < 0.05; (two-tailed Student's *t* test). **d** Subcellular localization of Rpp6907-7. Co-localization of Rpp6907-7::GFP with the mCherry marker. Bar = 20 μm. **e** The Rpp6907-7 and fusion proteins were transiently expressed in *N. benthamiana*. The numbers in parentheses indicate the numbers of leaves displaying cell death out of the total number of leaves infiltrated. Western blots probed with anti-GFP antibodies show the typical expression of all tested proteins. Expected protein size was 30 kDa for EV-GFP, 130 kDa for Rpp6907-7:GFP and Rpp6907-784:GFP. Modified ponceau staining is shown to indicate loading. **f** Y2H assay assessing the interaction of Rpp6907-7 CC domain. The constructs were co-transformed into yeast cells and grown on selective dropout medium as indicated.

Western blots probed with anti-HA and anti-cMyc antibodies show protein accumulation of all tested proteins in Y2H experiments. Unrelated protein SS12 (Glyma.14G199400) as negative controls to test the specificity of the interactions detected. Expected protein size was 18.6 kDa for AD, 40 kDa for CC-AD, 66 kDa for LRR-AD, 50 kDa for NB-ARC-AD, 43 kDa for CC-BD and S12-BD. Modified ponceau staining is shown to indicate loading. 3-amino-1,2,4-triazole (3-AT); DO Supplement -His/-Leu (DDO); DO Supplement -His/-Leu/-Met (TDO). **g** BiFC assay shows the interaction of Rpp6907-7 CC domain in *N. benthamiana*. Unrelated protein SS12 as negative controls to test the specificity of the interactions detected. Western blots probed with anti-HA and anti-Flag antibodies show protein accumulation of all tested proteins in BiFC experiments. Expected protein size was 33 kDa for Rpp6907-7CC-YFPC, 12 kDa for EV-YFPC, 39 kDa for Rpp6907-7CC-YFPN and SS12-YFPN. Modified ponceau staining is shown to indicate loading. All images are representative of results obtained from three independent experiments with similar results. Source data are provided as a Source Data file.

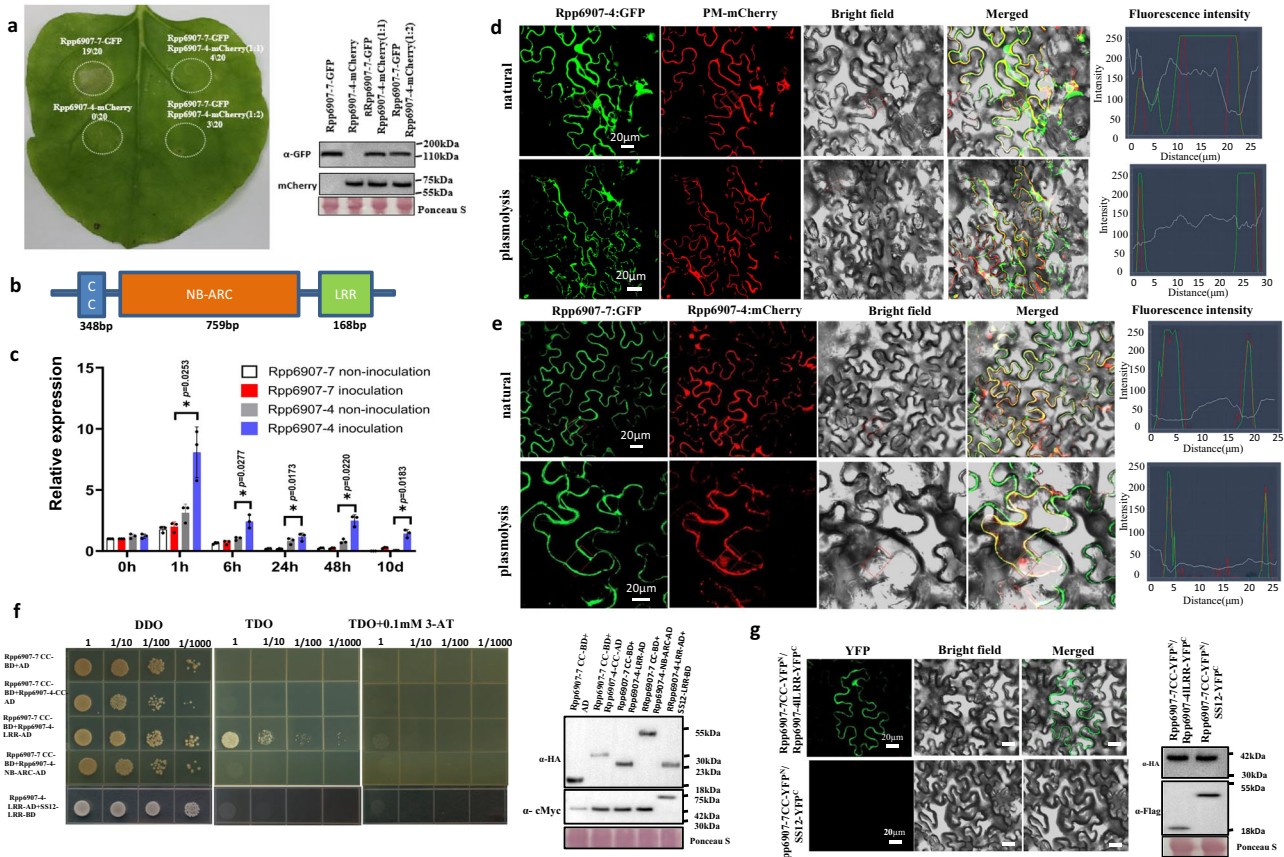

**Fig. 4 | Rpp6907-4 heterodimerizes interact with Rpp6907-7 and suppresses Rpp6907-7-mediated resistance. a** Rpp6907-4 suppresses HR induced by Rpp6907-7 in *N. benthamiana* leaves. The utilization of fusion GFP and mCherry were employed for the purpose of detecting the typical expression of the target protein. The numbers in parentheses indicate the numbers of leaves displaying cell death out of the total number of leaves infiltrated. Western blots probed with anti-HA and anti-mCherry show protein accumulation of all tested proteins. Expected protein size was 130 kDa for Rpp6907-7:GFP, 74 kDa for Rpp6907-4:mCherry. Modified ponceau staining is shown to indicate loading. **b** Structure of Rpp6907-4. **c** Relative expression of Rpp6907-7/Rpp6907-4 in SX6907 after inoculation and non-inoculation with SS4. Values are the mean ± SD (*n* = 3 repeats); *P < 0.05 (two-tailed Student's *t* test). **d** Subcellular localization of the fused Rpp6907-4: GFP in *N. benthamiana* leaf cells. Bar = 20 μm. **e** Localization patterns of Rpp6907-7 and Rpp6907-4. *N. benthamiana* leaves were transformed with combinations of constructs for the expression of Rpp6907-7: GFP, Rpp6907-4: mcherry. Bar = 20 μm.

**f** The CC domain of Rpp6907-7 and the LRR domain of Rpp6907-4 interact in yeast. The constructs were co-transformed into yeast cells and grown on selective dropout medium as indicated. Western blots probed with anti-HA and anti-cMyc show protein accumulation of all tested proteins. Expected protein size was 18.6 kDa for AD, 31 kDa for Rpp6907-4-CC-AD, 28 kDa for Rpp6907-4-LRR-AD, 48 kDa for Rpp6907-4-NB-ARC-AD, 43 kDa for Rpp6907-7 CC-BD, 69 kDa for S12-LRR-BD. Modified ponceau staining is shown to indicate loading. **g** The Rpp6907-7 CC domain and Rpp6907-4 LRR domain hetero-complexes were visualized by BiFC. Unrelated protein SS12 as negative controls to test the specificity of the interactions detected. Western blots probed with anti-HA and anti-Flag show protein accumulation of all tested proteins. Expected protein size was 39 kDa for Rpp6907-7CC-YFPN, 22 kDa for Rpp6907-4-LRR-YFPC, 39 kDa for SS12-YFPC. Modified ponceau staining is shown to indicate loading. All images are representative of results obtained from three independent experiments with similar results. Source data are provided as a Source Data file.

and growth presents an opportunity to uncover innovative insights that can be leveraged to cultivate plants with both high resistance and high yield. By disrupting critical regulatory links or nodes, this research can inform the development of novel strategies for identifying and characterizing resistance genes, which in turn can be used to enhance crop resilience against harmful microorganisms and ultimately bolster global food security.

## Methods

### Plant materials

The mapping populations were generated from an initial cross between Tianlong 1 and SX6907. SX6907 is a landrace obtained from Hubei Province, China that had an immune response to SS4, a *Phakopsora pachyrhizi* isolate from Fujian, China[11]. Tianlong 1 (high-yield variety developed by the Oil Crops Research Institute of the Chinese Academy of Agricultural Sciences) was used as the susceptible parent in this study. *N. benthamiana* plants were grown in a growth chamber at 24 °C with a 16 h light period.

### Construction, screening, and sequencing of a soybean bacterial artificial chromosome library

An SX6907 bacterial artificial chromosome (BAC) library was constructed by Nanjing Hong-Yuan Biotechnology Company Limited, Nanjing, Jiangsu, 210033, China. The genomic DNA was isolated from the leaves of SX6907, partially digested with BamHI, and inserted into the BAC cloning vector pSMART BAC to construct a BAC library with 125,000 clones with an average insert size of 110 kb. The BACs covering the *Rpp6907* locus were isolated using PCR-based markers SSR24 and SSR40 that co-segregate with Rpp6907 (Supplementary Data 6). Two positive BAC clones (BAC255 and BAC596) were identified that together spanned the entire interval between SSR24 and SSR40[45]. For each selected BAC clone, at least 300x Illumina paired-end short reads and 50x PacBio continuous long reads (CLR) were produced using an Illumina NovaSeq platform and PacBio Sequel platform, respectively. The two BAC sequences were assembled into one large contig of 179,816 bp using the Canu v1.9. The genomic sequence of the *Rpp6907* locus was annotated by using the gene prediction program Fgenesh (www.softberry.com) and was manually edited by a homology search against available databases on Phytozome (https://phytozome-next.jgi.doe.gov/). Seven complete NLR enncoding candidate gene sequences were identified in this contig; BAC255 carries six of the NLR encoding gene sequences (*R7*, *R6*, *R5*, *R4*, *R3*, and *R2*), and BAC596 carries two NLR encoding gene sequences (*R1*, *R2* and *R3*).

### Candidate gene identification

Overlapping primer pairs (Supplementary Data 6) designed along the entire length of the predicted genes were used to amplify the sequence of SX6907 (*Rpp6907* donor line) and various susceptible soybean germplasms. Amplified sequences were compared for nucleotide variations using DNAMAN software.

### Plasmid construction and soybean transformation

All PCR products used for cloning were generated using TransTaq® DNA Polymerase High Fidelity or TransStart® FastPfu Fly DNA Polymerase using the primers listed in Supplementary Data 6. The soybean cultivar Tianlong 1 and 06KG (susceptible to soybean rust isolates) were transformed with a genomic construct including *R7* with its native promoter. The construct used for transformation included 2525 bp upstream of the start codon, the complete coding region, and 2574 bp downstream of the stop codon. The fragment was amplified using gene-specific primers from BAC255, and the PCR products were inserted into the binary vector pTF101 to generate over-expression plasmids. To generate the *R7* RNAi constructs, the 265-bp fragment of *R7* was amplified and inserted as inverted repeats into the RNAi vector pB7GWIWG2(I) to generate hairpin RNAi constructs. The RNAi

constructs were introduced into the rust-resistant soybean cultivar SX6907 using the *Agrobacterium*-mediated cotyledonary node method[46].

For the yeast two-hybrid analysis, pGBKT7-BD and pGADT7-AD (Clontech) were used. For the creation of Gateway entry clones, pDONR/zeo or pGWC were used. For *Agrobacterium*-infiltration experiments, pEarleyGate100, pEarleyGate101-35S:: EYFP: HA, pEarleyGate103-35S::GFP:HA, pEarleyGate201-YN, pEarleyGate202-YC[47], pm-rkCD3-1008, pTRV1, pTRV2 and pTRV2-GFP were used.

### Pathogen assay

For inoculation, the fully expanded primary leaves of the soybean seedlings were collected, and each leaf used for inoculation was considered one replicate. The urediniospore suspension was adjusted to $10^5$ urediniospores per ml using Tween 20 (0.01% v/v). The detached leaves were placed in a plate (Φ15.0 × 2.0 cm) padded with a piece of wet filter paper. The upper surface of the leaf was in contact with the paper. Each leaf was inoculated with a suspension of urediniospore. The leaves were stored in the dark at 24 °C for the first night after inoculation, and then transferred to a growth chamber at 24 °C and 70% relative humidity (RH) under a 12/12-h photoperiod. Approximately 1–2 mL of water was added daily to keep the filter paper completely wet. Two weeks after inoculation, the leaves were scored for the presence of resistance response symptoms.

For plant spray inoculation, the urediniospore suspension was adjusted to $10^5$ urediniospores per mL using Tween 20 (0.01% v/v). The suspension was sprayed on the plants with a small watering can. Following inoculation, the plants were incubated for approximately 12 h at 24 °C in a dew chamber and later moved to a greenhouse maintained at 20–26 °C and 60% RH under a 12/12-h photoperiod for 14 days until rating symptoms. The SBR reactions were classified according to lesion type: IM type, >99% reduction in lesion counts per unit leaf area; RB type, reddish-brown lesions with no sporulation 14 days post infection and showed 60–71% reduction in lesion count per cm²; TAN type, the lesions had tan-colored lesions and a similar level of sporulation compared to susceptible controls 14 days after inoculation.

06KG transgenic events were evaluated with US and BR ASR populations: RTP1, 20BR-MT, 20BR-SUL, 20BR-Centro, Centro, 20BRS, 18BR1/2, 21BR-B1, 21BR-B3, 21BR-B4, 21BR-B5, 21BR-B6, 21BR-B7 and K8108. Tianlong 1 transgenic events were evaluated with China ASR isolate SS4.

### T₁ transgenic testing

Seeds from the first segregating generation (T₁) of events (Rpp6907.2, Rpp6907.6, 25911-05A, and RNAi-Rpp6907) were planted and maintained under growth chamber conditions for 17 d until the unrolled unifoliolate leaf growth stage. The plantlets were sampled for southern blotting to determine the transgene copy number and inoculated with a suspension of *Phakopsora pachyrhizi* spores. The inoculation was performed as described above with urediniospores collected from a susceptible variety and stored at −80 °C. To assess the effect of *Rpp6907-7*, the plants were scored qualitatively as IM, RB, or TAN type (Supplementary Data 2). Resistant response was scored 14 d after inoculation. To determine the effect of the gene, we compared the transgenic plants to the null plants from the same event; a separate analysis was performed for each zygosity level.

### Field trial, yield evaluation, and statistical analysis

Seeds from three independent transgenic plants from 25,913 individuals, one null segregant from 25,913 individuals, and one separate wild type lines were used in field trial. Each genotype was planted in a two-row plot with three replicates in a completely random design. Each plot is 1 meter (width) by 5 m (length) with approximately 220 plants per plot. Seed yield per plot was harvested, weighted, and recorded

statistical analysis was performed using a Student's *t* test or Tukey–Kramer test for multiple comparisons.

## RNA extraction and quantitative RT-PCR (qPCR) analysis

Total RNA was extracted from different soybean tissues using TRIzol reagent and treated with RNase-free DNase I according to the manufacturer's protocol (Invitrogen). cDNA was synthesized using a HiScript® II Q RT SuperMix for qPCR (+gDNA wiper) kit (Vazyme). qPCR was performed using SYBR Green (Bio-Rad) with the CFX Connect Real-Time System (BIO-RAD; 45 cycles of 95 °C for 10 s and 60 °C for 30 s, with signal acquisition at the end of each amplification cycle). $\beta$-actin was used as an internal control, and the $2^{-\Delta\Delta Ct}$ method was used to calculate relative gene expression. Three samples were collected from each tissue type, and each sample was collected from at least five individuals. RNA expression for each sample was detected in three technical replicates. The primers for qRT-PCR are listed in Supplementary Data 6.

## Tissue sampling for determination of expression patterns

SX6907 seeds were sterilized, sown in sterile soil, and grown on a 16-h light (28 °C) to 8-h dark (25 °C) cycle. For inoculation, the fully expanded primary leaves of the soybean seedlings were collected. To analyze the induced expression pattern, inoculated and uninoculated leaf samples were collected at 0 h, 1 h, 6 h, 24 h, 48 h, and 10 d for qPCR. Three biological replicates were included for each infection time point. For seedling sample, the leaf, shoot, and stipe were sampled 14 days after sowing. For adult plant sampling, plants were grown in the field at the ORIC-CAAS experimental station in Wuhan, China. Different tissues were sampled at the pod bearing stage. Each sample contained tissue from at least five individuals..

## Phylogenetic analysis of CNL proteins

We downloaded all annotated plant CNL protein sequences from the NCBI database. Phylogenetic trees in this study were constructed using the minimum evolution method in MEGA6.0[48].

## Subcellular localization

The coding sequences of *Rpp6907-7* and *Rpp6907-4* were amplified using the cDNA clone as the template. The PCR product was inserted into the pEarleyGate101, CD3-1008 (mCherry), or pEarleyGate103 vector to create open reading frame (ORF)-encoding fluorescent-fused proteins driven by the 35S promoter. All constructs were transferred into *Agrobacterium tumefaciens* strain GV3101. The obtained Agrobacterium strains were used to infiltrate *N. benthamiana* leaves. Transfected *N. benthamiana* leaves were assayed for fluorescence with a Zeiss confocal microscope[35].

## Yeast two-hybrid assays

pGBKT7(BD) or pGADT7(AD) constructs used in our study were transformed into the yeast strain Y2H gold (Clontech). Co-transformants were plated on synthetic medium lacking leucine, tryptophan, adenine, and histidine at 28 °C for 3 days. Experimental procedures for screening and plasmid isolation were performed according to the manufacturer's instructions (Clontech).

## Bimolecular fluorescence complementation analysis

For the in vitro bimolecular fluorescence complementation assay in *N. benthamiana*, genes of interest were cloned into the BiFC vectors pEarleyGate201-YN and pEarleyGate202-YC. *A. tumefaciens* strain GV3101 harboring YN and YC constructs were mixed in a (1:1):1 ratio with the *A. tumefaciens* p19 strain prior to infiltration into *N. benthamiana* plants. The transient expression in the transfected *N. benthamiana* leaves were assayed for fluorescence with a Zeiss confocal microscope[35].

## Transient protein expression and cell death assays in *N. benthamiana*

For *Agrobacterium*-mediated *N. benthamiana* leaf transformations, transformed GV3101 strains were grown in Luria-Bertani liquid medium containing 50 mg/mL rifampicin, 15 mg/mL gentamycin, and 25 mg/mL kanamycin at 28 °C for 24 h. Bacteria were harvested by centrifugation, resuspended in infiltration medium (10 mM MES pH 5.6, 10 mM MgCl$_2$, 150 μM acetosyringone) to an OD600 of 1.0, and incubated for 2 h at room temperature before leaf infiltration. The infiltrated plants were incubated in growth chambers under controlled conditions for bimolecular fluorescence complementation (BiFC) and cell death assays. For documentation of cell death, leaves were photographed 3–5 days after infiltration. Epidermal cell layers of *N. benthamiana* leaves were assayed for fluorescence with a Zeiss confocal microscope 1–2 days after infiltration.

## Characterization of site-directed mutants

According to results from the multiple sequence alignment of *Rpp6907-7* in 17 susceptible soybean cultivars, the resistant *Rpp6907-7* differs in two short regions with a 15-bp insertion, a 9-bp deletion, a 6-bp deletion, a 3-bp deletion, and 36 single nucleotide polymorphisms (SNPs). Site-specific mutations were used to identify functional regions of genes. The Fast Mutagenesis System (TransGen Biotech) was used for site-directed mutagenesis. The mutation primers were designed according to product specifications. We designed 35 pairs of mutation primers (Supplementary Data 4). All mutant plasmids were transferred into the *A. tumefaciens* strain EHA105. To test which amino acid positions were associated with resistance in Rpp6907, we used the obtained *Agrobacterium* strains to infiltrate *N. benthamiana* leaves for observing hypersensitive response (HR)-like cell death.

## Protein extraction and western blot analysis

To extract total protein, yeast strains were grown overnight in 10 mL liquid medium without leucine and tryptophan at 30 °C. After centrifugation for 5 min at 2500 × g, cells were washed with 1 mM EDTA. After resuspending the pellet in 200 μL of 0.15 M NaOH, it was incubated on ice for 10 min. The sample was then incubated on ice for 1 hour with 200 μL of 50% trichloroacetic acid (TCA). 200 μL acetone was used to wash the pellet after centrifugation at 13,000 × g for 20 min. After resuspending the pellet in 200 μL 5% SDS, 200 μL of 1x Laemmli sample buffer was added. With 1 M TRIS pH 10, the pH of the sample was adjusted to 4.6, and the sample was incubated at 37 °C for 15 min. The supernatant was collected after centrifugation for 15 minutes at 13,000 × g.

We collected two leaf discs per leaf from four independent *N. benthamiana* leaves 2-3 dpi for total protein extraction. We immediately froze the tissue using liquid nitrogen and ground it with a mortar and pestle. The sample was heated in 150 μL SDS sample buffer for 10 min at 95 °C. We centrifuged the samples twice for 5 min at 16,000 × g and collected the supernatant.

Total protein samples were separated on an SDS-PAGE gel and transferred to a Immobilon-PSQ PVDF membrane (Merck-Millipore). Membranes were probed with mouse monoclonal [HA.C5] to HA tag (Abcam, Cat#ab18181), 1:1000; mouse monoclonal [9E10] to Myc tag (Abcam, Cat#ab32), 1:1000; mouse anti GFP-Tag mAb (Abclonal, Cat#AE012), 1:2000; HRP Anti-DDDDK tag (Binds to FLAG® tag sequence) (ABclonal, Cat#AE005), 1:1000; mouse anti mCherry-Tag mAb (Abclonal, Cat#AE002), 1:5000; HRP conjugated Goat Anti-Mouse IgG (H + L) secondary antibody (Abbkine, Cat#A21010), 1:10000. Antibodies were detected by HRP activity on western Bright ECL HRP substrate (servicebio) using the Automatic chemiluminescence image analysis system (Tanon 5200).

## Virus-induced gene silencing

For virus-induced gene silencing (VIGS) assays, pTRV1, pTRV2:Rpp6907-4, or pTRV2:GFP plasmid constructs were introduced into *Agrobacterium tumefaciens* EHA105. The cultured Agrobacterium cells were harvested and resuspended in infiltration buffer (40 mg/L AS, 150 mg/L DTT, 400 mg/L Cys, 1.0 mg/L 6-BA, pH5.4) to an OD600 of 0.5–1.0. Cotyledon note removed embryo root from 10 to 24 h germinating seeds were used as explants. Explants were put into *Agrobacterium* suspension for 20 min, then explants were transferred to the plates containing semi-solid co-cultivation medium (1/10 MS, containing 40 mg/L AS, 150 mg/L DTT, 400 mg/L Cys, 1.0 mg/L 6-BA, 3 g/L agar, pH5.4) for 5 days culture in dark. After co-cultivation, the explants were transfered to medium (MS with 0.5 mg/L 6-BA, 8 g/L agar, pH5.8) for shoot development. In 2-3 weeks, shoots with root can be transplanted into pots, growth condition for plantlets were temperature at 25 °C, 16/8 h photoperiod at a light level of 150 μmol photons m$^{-2}$ s$^{-1}$ and 55% relative humidity. Systemic silencing may occur 4–5 weeks post-agroinfiltration. Leaves from plants can be collected for rust pathogen inoculation, and RNA extraction.

## Reporting summary

Further information on research design is available in the Nature Portfolio Reporting Summary linked to this article.

## Data availability

The *Rpp6907* locus genomic DNA sequences from SX6907 is available at NCBI under accession PP508376. Data supporting the findings of this work are provided in the paper and its Supplementary Information file. The genetic materials supporting the findings of current study are available from the corresponding authors upon request. Source data are provided with this paper.

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

## Acknowledgements
We thank C. Li, T. Liebler, L. Haislip, M. Terry, D. McNamara for plant and seed management, C. Fan for Taqman, and R.C.M. Lunny for sampling. We thank M. Murray, S. Patton for Vector QC, Q. Que and L. Shi for laboratory and greenhouse space and personnel support, and B. Vernooij and E. Chen for project support. Thanks to J. Ho, S. Miles, X. Xue, K. Jones, K. Williamson, and S. Kanjilal for licensing and guidance on intellectual property. This research was supported by China Agriculture Research System of MOF and MARA (CARS-04-PS10) to X.Z., the Agricultural Science and Technology Innovation Program of Chinese Academy of Agricultural Sciences(CAAS-ASTIP-2022-OCRI) to H.C. Basic research fund of Oil Crop Research Institute, Chinese Academy of Agricultural Sciences (1610172022002) to Q.H.

## Author contributions
Q.H., Q.L., Z.S. and X.Z. conceived the project; Q.H., H.Y., S.C., C.Z., D.C., Y.H., L.Q., X.S., X.W., Q.L., H.C. and Z.S. designed the research; Q.H., H.Y., S.C., L.C., W.G., L.D., Y.Q., H.Z., E.B., W.W., Y.J., C.M., J.D. and R.C. performed most of the experiments and analyzed the data; Q.H., H.Y., S.C., S.Y., Z.Y. and G.T. visualized the data; X.Z. and H.C. provided funding; X.Z., Q.L. and Z.S. supervised the project; Q.H. and Z.S. wrote the original draft; Q.H., Z.S., Q.L. and X.Z. edited the manuscript.

## Competing interests
Two patents covering identification, selection, and/or production of soybean plants or germplasms resistant to Asian soybean rust using markers, genes and chromosomal intervals derived from *Glycine max* cultivar SX6907 were filed on Dec 4, 2020 (CN112239491 and WO2021000878). Inventors of the patents are Z.S., Q.H., H.C., Y.Y., C.Z., L.C., S.Y., D.C., W.G., X.Z., S.C., Z.Y., D.Q, X.Z., Q.L., B.B., S.G. Other authors don't claim competing interests.
