## [Peer Review File · Nature Communications]

An atypical NLR pair confers Asian soybean rust resistance in soybeanReviewers' Comments:

Reviewer #1:

Remarks to the Author:

This manuscript by Hao et al. reports the first case of Asin Soybean Rust (ASR) resistance gene identified from soybean. The authors use resistant soybean landraces to identify Rpp6907-7 and Rpp6907-4. Despite their distances in the position of genome, which makes it atypical paired NLR, the authors report the two genes function together. Overall, this manuscript reports an important discovery that is much needed for ASR resistance in soybean crops. However, much improvement is needed with analysis and interpretation of experimental data. Absence of statistical analysis, control experiments, and lack of transparent data visualization is concerning. All results of statistical analysis must be shown in the figures or the figure legends, and it is suggested that all data points are shown in the graphs. For negative results of Y2H, BiFC, and HR assay data, corresponding proof that the result is negative despite normal protein expression must be shown.

Major comments

Fig S1: Why are southern blot patterns different between gDNA of SX6907 and BAC clone derived from the same gDNA? Also, Williams82 or Tianlong1 gDNA as negative control is needed to illustrate the additional NLR genes identified from SX6907.

For Fig S1B-C, a different soybean NLR should be used as outgroup. With CcRpp1, the results obtained by the authors are inevitable. If CcRpp1 is to be used as outgroup, all NLRs from soybean should be used for phylogeny comparison. Also, NB domains should be used for phylogeny analysis, as these are the most conserved domains of the NLR and makes the comparison more robust.

Fig S2: the table with SNPs are hard to understand. What do the numbers across the top mean? What does R and S1-13 mean? In Line 102 'solely R7 exhibited specificity towards SX6907' does not align with having R7 homologs identified from susceptible varieties. Please clarify.

Does inoculation have effect on Rpp6907-7 expression? Add non-inoculated sample to ensure inoculation itself does not have effect on expression. Statistical analysis is required. Same control samples should be included for Fig 4C.

There is discordance between EGFP localization data and BiFC data. Have the authors tried plasmolysis to see that the GFP localization is indeed at the plasma membrane? The plasma membrane marker seems insufficient to conclude plasma membrane localization, as the localization data observed seems to be cytosolic. The BiFC data is pixelated and thus unclear, but it seems that the interaction of full-length Rpp6907-7 and CC domain of Rpp6907-7 occurs throughout the cell. This creates contradiction between EGFP localization and BiFC interaction data. Also, Rpp6907-4 is detected as nucleocytoplasmic protein. Upon co-expression with Rpp6907-4, Rpp6907-7 localization is also detected in nucleus, and not just in the PM. Is this change in localization of Rpp6907-7 only observed in the presence of Rpp6907-4? This will be very interesting, as it would suggest that the sensor/repressor NLR re-localize the executor NLR.

Fig S5, there is no information of which amino acids have been mutated. Table S3 is just primer information, each amino acid substitution should be described in actual figure.

Does RPP6907-7 Q84 lose self association of CC domains?

Fig S4, statistical analysis for occurrence of HR of the individual domains is requested. The CC domain seems to be showing HR, albeit weakly. Also, have the authors tested tagging Rpp6907-7 CC domain with GFP and testing for HR? GFP can dimerize/oligomerize and can often lead to HR induction that could not be observed in none-tagged versions.

Protein expression of all negative HR, Y2H, and BiFC data is requested. Whether or not HR is suppressed by co-expression or absence of expression cannot be excluded without this data.

Statistical analysis of HR assay is also needed. This includes but not limited to Fig 3, 4, S7, 8.

Evidence is needed for Line 275-276. Phylogeny tree of Rpp6907-4 and the integrated decoys of these domains should be shown. Is Rpp6907-4 clade similar to the MIC clade identified in monocots?

Additional data of testing the other R1-R6 NLRs for suppression of R7 should be included, along with proof of expression of the cloned constructs.

Minor comments

Line59 All references including html links should be referenced and cited.

Line87 Is the 56.77kb additionally found besides the interval found between susceptible accession Williams82 and SX6907? The use of 'Furthermore' suggests there are actually ~100kb interval?

Tobacco should all be universally written as 'Nicotiana benthamiana'.

Line 206-208 is unnecessary; it is already highlighted in multiple review articles about function of each domains.

Line218 Structural analysis show that for CC-NLRs, oligomerization is needed, not dimerization.

Remove Line263. At least a temporal analysis is needed to suggest the claim of Line263.

Line268 BiFC data is not evidence for oligomerization, it suggests dimerization, not any more than that.

Reference list is incomplete. Refereneces 46 and 47 could not be found in main text, and Kasmi et al. (36) could not be found in the reference list. Original or first publications regarding a discovery should be referenced, rather than the latest publication that developed original discovery.

Reviewer #2:

Remarks to the Author:

In this study, Hoa and colleagues describe the identification of the Rpp6907-7 NLR encoding gene from Chinese soybean SX6907 that confers a broad spectrum resistance to Asian Soybean Rust (ASR) caused by a biotrophic fungus (*Phakopsora pachyrhizi*). Authors present interesting data concerning the map-based cloning and the functional characterization of Rpp6907-7. Involvement of Rpp6907-7 in resistance against ASR was investigated through genetic complementation of different susceptible lines and the data presented that are presented in this part seem rather convincing (Fig 1, FigS1-S3, Table S1-S2). To further confirm the role of Rpp6907-7 in resistance to ASR, authors performed RNAi to silence its expression in the resistant germplasm. It is highly regrettable that the authors did not check the extent to which R7 expression levels were affected in the selected RNAi transgenic lines, nor did they omit to include control RNAi constructs. As a result, the conclusions drawn by the authors should be viewed with caution.

Using various approaches (confocal microscopy, transient assay in *N. benthamiana*, split-YFP, yeast two-hybrid), the authors have attempted to study the mode of action of this NLR receptor from various angles (subcellular localization, autoactivity, oligomerization, intramolecular interactions). Some of these aspects related to NLR biology can quite challenging. Unfortunately, in this study, in most of the experiments presented, appropriate controls that are necessary to support the authors' conclusions are missing. For instance, in the split-YFP assay, the proper expression of the different proteins either fused to YFPc or YFPn should be checked by immunoblot. In addition, unrelated proteins showing the same subcellular localization should be included as negative controls to test the specificity of the interactions detected (Fig 3F-G, 4G). In line with this, all transient expression experiments performed in *N. benthamiana* should include immunoblots to verify that the different proteins of interest are properly expressed (Fig 3E, 4A, S4). In the Y2H experiments, important controls are also missing. Each bait (or prey) must be co-expressed with a negative prey (or bait) construct to ensure that the bait/prey constructs of interest do not exhibit self-activation (and the expression of the different AD/BD fusion proteins should be checked by immunoblot). Therefore, for all these reasons, I think that all the authors' conclusions regarding the self-association of Rpp6907-7 by its CC domain are not supported by sufficiently convincing data.

Given that expression of Rpp6907-7 triggers a cell death response in *N. benthamiana*, authors hypothesized that Rpp6907-7 autoactivity could be regulated through intermolecular interactions with a second NLR (the sensor) that would function in pair with Rpp6907-7 (the executor/helper). This hypothesis stems from the mode of action of several pairs of NLR that have been described to date. And indeed, this hypothesis is attractive and deserved to be tested. Authors tested all the other genes in the Rpp6907 locus (R1 to R6) by coexpressing each of them with Rpp6907-7. Here, it would have

been very informative to show these data (NLR-triggered cell death response in *N. benthamiana* and immunoblot assays to verify the proper expression of the different epitope-tagged NLR candidates). Interestingly, authors found that co-expression of Rpp6907-7 with Rpp6907-4 led to the inhibition of Rpp6907-triggered cell death, suggesting that Rpp6907-4 might function in pair with Rpp6907-7 to maintain the receptor complex in an inactive state. As mentioned above, I have the same comments to make to the authors regarding their interpretation of the presented data. In Fig4A, please show by immunoblots that both NLR proteins are properly expressed and migrates at the expected size. Same in Fig 4D and 4E. In Fig4F-4G, important negative controls and immunoblots are missing (see my previous comments for Fig3).

Overall, I think that involvement of Rpp6907-4 in suppression of Rpp6907-7-mediated resistance should be supported with some other (genetic?) data. According to their model, in soybean, does Rpp6907-mediated resistance require Rpp6907-4? This could/should be investigated through RNAi to specifically silence Rpp6907-4 expression and check if it does compromise or not Rpp6907-7-mediated resistance.

Reviewer #3:

Remarks to the Author:

Soybean rust is an aggressive disease of soybeans and other legumes caused the fungal pathogen *Phakopsora pachyrhizi*. The geographical distribution of *P. pachyrhizi* includes all major soybean production areas, and it is particularly problematic in several South American countries, most notably the world's largest soybean producer - Brazil. Due to a lack of durable resistance, the disease is primarily controlled by fungicide applications, and there is growing concern that several fungicides are becoming less efficacious. To date, at least 7 loci that confer resistance to *P. pachyrhizi* (Rpp) have been identified and mapped. Candidates for two of these genes (Rpp1 and Rpp4) have been identified and characterized using a loss-of-function approach using virus-induced gene silencing. Significantly, with the exception ccRpp1 identified in *Cajanus cajan* (pigeon pea), none of the known Rpp genes (Rpp1 – Rpp7) have been cloned. The work described by Hao et al. therefore represents a significant advance in our understanding of soybean rust resistance. In this manuscript the authors describe the identification and cloning of Rpp6907 obtained from a Chinese soybean landrace (SX6907). Candidate resistance genes were previously mapped to a region ~112-kb region of chromosome 18 in SX6907, which is a distinct locus from Rpp1, Rpp1b, and Rpp4 which also map to chromosome 18. By sequencing two bacterial artificial chromosomes spanning the Rpp6907 locus, seven candidate genes (R1 – R7) were identified and cloned into a susceptible soybean line. R7, or Rpp6907-7, was found to confer resistance (complete immunity) to the soybean rust isolate SS4 in homozygous plants. Hemizygous plants exhibited reddish-brown lesions, indicating that Rpp6907-7 is an incomplete dominant gene. Rpp6907-7 RNAi constructs in SX6907 resulted in susceptibility. The expression of Rpp6907-7 does not cause any changes in morphology and does not impact yield. A survey of 30 diverse soybean genomes indicates variability at the Rpp6907.

Characterization of Rpp6907-7 indicates that gene is not induced following challenge with *P. pachyrhizi* and is localized to the plasma membrane. Interestingly, expression of Rpp6907-7 causes localized cell death when expressed in *N. benthamiana* in the absence of any effector or other avirulence trigger. The authors utilized this to further characterized the expressed protein. The authors also exploit this observation to identify a second gene, Rpp6907-4, that appears to block the cell death induced by expression of Rpp6907-7. The authors propose that Rpp6907-7 and Rpp6907-4 work in tandem, although the precise mechanisms are not fully understood.

For the most part the data is clear and appropriate controls are included. However, this is not the case for the studies involving expression in *N. benthamiana*, where Western blot data is needed to confirm the expression of truncated proteins and mutant proteins that do not elicit cell death. Also, virtually no description and no sequence is provided for Rpp6907-4. Importantly, the authors should also confirm

that Rpp6907-4 from the susceptible lines used for Rpp6907-7 expression (Tianlong1 and 06KG) can also suppress the cell death induced by Rpp6907-7.

Line 29 – The assertion that only SX6907 is the only soybean cultivar that has an immune response (IR) to ASR needs to be clarified or qualified. Rpp1 from PI 200492 is known to confer the IR to select isolates of *P. pachyrhizi*. In the reference cited (#3) only a single isolate was used to confirm the IR in SX6907.

Line 35 – the description of the roll of Rpp6907-4 as an inactive signaling executer is not clear to me, and the mention of non-canonical domains is equally puzzling.

Line 59 – are there food safety concerns associated with soybean rust?

Line 60 – it would be better to first discuss the resistance genes from soybean (Rpp1 – 7) before introducing ccRpp1.

Line 75 – The claim of broad-spectrum resistance conferred by SX6907 needs to be supported – the reference cited used only one isolate.

Line 89 – The description of the location of R1 – R6 and R7 could be simplified by indicating the genes flanking R1 – R6 and R7

Line 98 – Referring to R1- R6 and R7 as “intact” is confusing. Perhaps just drop “intact” as the length of the predicted proteins is stated. Likewise, provide the predate length of R4 instead of just saying it is truncated.

Line 117 change heterozygous to hemizygous (correct in figure)

Line 134 – provide more detail about the *P. pachyrhizi* populations used. I did not see this in the materials and methods.

Line 192... contains a single 2,571 bp exon and no introns

Line 247 It’s not clear what is meant by “in soybean cells that are absent from *N. benthamiana*”

Line 275 – does this imply that Rpp6907-4 is involved in the recognition event?

Dear prof. An and Reviewers,

Thank you for your letter and the reviewers' comments concerning our manuscript entitled "A novel atypical NLR pair confers Asian Soybean Rust in soybean" (ID:NCOMMS-23-37156A). Thanks very much for taking your time to review this manuscript. I really appreciate all your comments and suggestions! Please find my itemized responses in below and my revisions/corrections in the re-submitted files.

Thanks again!

COMMENTS TO THE AUTHOR:

Reviewer #1:

Q1. Fig S1: Why are southern blot patterns different between gDNA of SX6907 and BAC clone derived from the same gDNA? Also, Williams82 or Tianlong1 gDNA as negative control is needed to illustrate the additional NLR genes identified from SX6907.

Response to the questions one by one: Thank you for your suggestion. As suggested by reviewer, we have added Williams82 and Tianlong1 gDNA as negative control for southern blot of Fig. S1A. Because of two positive BAC clones were identified and assembled encompassed the Rpp6907 locus, BAC255 carries six of the NLR gene sequences (R7, R6, R5, R4, R3, and R2), and BAC596 carries two NLR gene sequences (R1, R2 and R3), so southern blot patterns different between gDNA of SX6907 and BAC clone.

Q2. For Fig S1B-C, a different soybean NLR should be used as outgroup. With CcRpp1, the results obtained by the authors are inevitable. If CcRpp1 is to be used as outgroup, all NLRs from soybean should be used for phylogeny comparison. Also, NB domains should be used for phylogeny analysis, as these are the most conserved domains of the NLR and makes the comparison more robust.

Response to the questions one by one: We are grateful for the suggestion. As suggested by reviewer, we have added different soybean NLR as outgroup Fig S1B-C. We also used the NB domain of these genes for phylogeny analysis in Fig S1B.

Q3. Fig S2: the table with SNPs are hard to understand. What do the numbers across the top mean? What does R and S1-13 mean? In Line 102 'solely R7 exhibited specificity towards SX6907' does not align with having R7 homologs identified from susceptible varieties. Please clarify.

Response to the questions one by one: Thank you for your careful review. According to the reviewer's comment, we have modified Fig S2 to present the SNP in a clearer way. A multiple sequence alignment was performed on the gene with the highest sequence similarity to R7 in 13 soybean varieties to examine the mutations present. We corrected the discrepancy between the two accounts (Line 102 'solely R7 exhibited specificity towards SX6907' does not align with having R7 homologs identified from susceptible varieties), "R7 homologs identified from susceptible varieties" should be expressed as "R7 highest sequence similarity genes identified from susceptible varieties".

Q4. Does inoculation have effect on Rpp6907-7 expression? Add non-inoculated sample to ensure inoculation itself does not have effect on expression. Statistical analysis is required. Same control samples should be included for Fig 4C.

Response to the questions one by one: Thank you for your suggestion. The patterns of Rpp6907-7 expression were examined both non-inoculated and inoculation with SS4 was showed in Fig.3C, revealing that the expression of Rpp6907-7 was not affected by inoculation two days before. The results of statistical analysis have been shown in the figures. Fig.3C and Fig 4C data were normalized to the non-inoculation Rpp6907-7 sample at 0 hours.

Q5. There is discordance between EGFP localization data and BiFC data. Have the authors tried plasmolysis to see that the GFP localization is indeed at the plasma membrane? The plasma membrane marker seems insufficient to conclude plasma membrane localization, as the localization data observed seems to be cytosolic. The BiFC data is pixelated and thus unclear, but it seems that the interaction of full-length Rpp6907-7 and CC domain of Rpp6907-7 occurs throughout the cell. This creates contradiction between EGFP localization and BiFC interaction data. Also, Rpp6907-4 is detected as nucleocytoplasmic protein. Upon co-expression with Rpp6907-4, Rpp6907-7 localization is also detected in nucleus, and not just in the PM. Is this change in localization of Rpp6907-7 only observed in the presence of Rpp6907-4? This will be very interesting, as it would suggest that the sensor/repressor NLR re-localize the executor NLR.

Response to the questions one by one: Thank you for your suggestion. As suggested by reviewer, we have further demonstrated the subcellular localization of Rpp6907-7 by plasmolysis, and we found that the Rpp6907-7 GFP localization is indeed at the plasma membrane (shown in Fig.3D). When Rpp6907-7 and Rpp6907-4 were co-expressed, their localization changed and they could be co-localized on the plasma membrane (shown in Fig.4E).

Q6. Fig S5, there is no information of which amino acids have been mutated. Table S3 is just primer information, each amino acid substitution should be described in actual figure.

Response to the questions one by one: Thank you for your suggestion. As suggested by reviewer, each amino acid substitution have been described in Fig S5.

Q7. Does RPP6907-7 Q84 lose self association of CC domains?

Response to the questions one by one: Thank you for your question. we found Rpp6907-7 CC domains self-associates wasn't lost in 84 (Q to R) mutation. Y2H and BiFc results were shown in Fig.S7.

Q8. Fig S4, statistical analysis for occurrence of HR of the individual domains is requested. The CC domain seems to be showing HR, albeit weakly. Also, have the authors tested tagging Rpp6907-7 CC domain with GFP and testing for HR? GFP can dimerize/oligomerize and can often lead to HR induction that could not be observed in none-tagged versions

Response to the questions one by one: Thank you for your suggestion. All the HR experiments were repeated multiple times. The representative images from a single replicate of three independent experiments are shown. As suggested by reviewer,

We've done statistical analysis for the occurrence of HR and annotated it in Fig. S4 or other HR-related results. Rpp6907-7 different domains with GFP have been test for HR(shown in Fig.S4).

Q9. Protein expression of all negative HR, Y2H, and BiFC data is requested (HR, Y2H, and BiFC. Whether or not HR is suppressed by co-expression or absence of expression cannot be excluded without this data. Statistical analysis of HR assay is also needed. This includes but not limited to Fig 3, 4, S7, 8.

Response to the questions one by one: We are extremely grateful to reviewer for pointing out this problem. We have supplemented all the protein expression data of all negative HR, Y2H, and BiFC by WB or Immunofluorescence(shown in Fig.3, 4, S4, S5, S7, S8 and S11). We've done statistical analysis for all HR assay and annotated it in Fig.3, 4, S4, S5, S8 and S11.

Q10. Evidence is needed for Line275-276. Phylogeny tree of Rpp6907-4 and the integrated decoys of these domains should be shown. Is Rpp6907-4 clade similar to the MIC clade identified in monocots?

Response to the questions one by one: Thank you for your suggestion. The phylogeny tree of decoy NLR protein from all studied functionally coupled pairs was constructed(FigS10), including MIC clade members. Since Rpp6907-4 does not have an ID domain, we hypothesized that it may perform a different function than other members. Our understanding of Rpp6907-4 will facilitate the fine-tuning of immune activation of new NLR-ID fusions. These are just some of my ideas, there is no direct evidence yet. This part of the work will be done next.

Q11. Additional data of testing the other R1-R6 NLRs for suppression of R7 should be included, along with proof of expression of the cloned constructs.

Response to the questions one by one: Thank you for your suggestion. We have included data of other R1-R6 NLRs for suppression of R7 (shown in Fig.S8).

Minor comments

Q12. Line59 All references including html links should be referenced and cited.

Response to the questions one by one: We are extremely grateful to reviewer for pointing out this problem. All references including html links have been referenced and cited.

Q13. Line87 Is the 56.77kb additionally found besides the interval found between susceptible accession Williams82 and SX6907? The use of 'Furthermore' suggests there are actually ~100kb interval?

Response to the questions one by one: We are extremely grateful to reviewer for pointing out this problem. I have been corrected for my inaccuracy of Line87. Change 'Furthermore' to 'summary'.

Q14. Tobacco should all be universally written as 'Nicotiana benthamiana'

Response to the questions one by one: We are extremely grateful to reviewer for pointing out this problem. The writing of tobacco was unified.

Q15. Line 206-208 is unnecessary; it is already highlighted in multiple review articles about function of each domains

Response to the questions one by one: Thank you for your suggestion. We have

removed this part.

Q16. Line218 Structural analysis show that for CC-NLRs, oligomerization is needed, not dimerization.

Response to the questions one by one: Thank you for your suggestion. We have corrected the wording.

Q17. Remove Line263. At least a temporal analysis is needed to suggest the claim of Line263.

Response to the questions one by one: Thank you for your suggestion. We have remove Line263.

Q18. Line268 BiFC data is not evidence for oligomerization, it suggests dimerization, not any more than that.

Response to the questions one by one: We are extremely grateful to reviewer for pointing out this problem. We have corrected the wording.

Q19. Reference list is incomplete. Refereneces 46 and 47 could not be found in main text, and Kasmi et al. (36) could not be found in the reference list. Original or first publications regarding a discovery should be referenced, rather than the latest publication that developed original discovery.

Response to the questions one by one: We are extremely grateful to reviewer for pointing out this problem. We have revised the references.

Reviewer #2:

Q1. In this study, Hao and colleagues describe the identification of the Rpp6907-7 NLR encoding gene from Chinese soybean SX6907 that confers a broad spectrum resistance to Asian Soybean Rust (ASR) caused by a biotrophic fungus (*Phakopsora pachyrhizi*). Authors present interesting data concerning the map-based cloning and the functional characterization of Rpp6907-7. Involvement of Rpp6907-7 in resistance against ASR was investigated through genetic complementation of different susceptible lines and the data presented that are presented in this part seem rather convincing (Fig 1, FigS1-S3, Table S1-S2). To further confirm the role of Rpp6907-7 in resistance to ASR, authors performed RNAi to silence its expression in the resistant germplasm. It is highly regrettable that the authors did not check the extent to which R7 expression levels were affected in the selected RNAi transgenic lines, nor did they omit to include control RNAi constructs. As a result, the conclusions drawn by the authors should be viewed with caution.

Response to the questions one by one: Thank you for your summary. We are extremely grateful to reviewer for pointing out the problem about RNAi. We have done relevant experiments during the initial phase, but due to our negligence, we did not present relevant results in the Fig of the article. We have supplemented the data of R7 expression levels in RNAi transgenic lines(shown in Fig S3B). The phenotype of control RNAi constructs(RNAI-GFP) has been shown in Fig 1B-f.

Q2. Using various approaches (confocal microscopy, transient assay in *N. benthamiana*, split-YFP, yeast two-hybrid), the authors have attempted to study the mode of action of this NLR receptor from various angles (subcellular localization, autoactivity, oligomerization, intramolecular interactions). Some of these aspects

related to NLR biology can quite challenging. Unfortunately, in this study, in most of the experiments presented, appropriate controls that are necessary to support the authors' conclusions are missing. For instance, in the split-YFP assay, the proper expression of the different proteins either fused to YFPc or YFPn should be checked by immunoblot. In addition, unrelated proteins showing the same subcellular localization should be included as negative controls to test the specificity of the interactions detected (Fig 3F-G, 4G). In line with this, all transient expression experiments performed in *N. benthamiana* should include immunoblots to verify that the different proteins of interest are properly expressed (Fig 3E, 4A, S4). In the Y2H experiments, important controls are also missing. Each bait (or prey) must be co-expressed with a negative prey (or bait) construct to ensure that the bait/prey constructs of interest do not exhibit self-activation (and the expression of the different AD/BD fusion proteins should be checked by immunoblot). Therefore, for all these reasons, I think that all the authors' conclusions regarding the self-association of Rpp6907-7 by its CC domain are not supported by sufficiently convincing data.

Response to the questions one by one: Thank you for your suggestion. As suggested by reviewer, we have supplemented all the protein expression data of all negative HR, Y2H, and BiFC by WB or Immunofluorescence (shown in Fig.3, 4, S4, S5, S7, S8 and S11). We've done statistical analysis for all HR assay and annotated it in Fig.3, 4, S4, S5, S8 and S11. Unrelated protein SS12 (Glyma.18G283200 1-182 CC domain) as negative controls to test the specificity of the interactions detected in Fig 3F-G and 4G. controls and expression of the different AD/BD fusion proteins have been supplemented in Fig.3, 4, Fig S7.

Q3. Given that expression of Rpp6907-7 triggers a cell death response in *N. benthamiana*, authors hypothesized that Rpp6907-7 autoactivity could be regulated through intermolecular interactions with a second NLR (the sensor) that would function in pair with Rpp6907-7 (the executor/helper). This hypothesis stems from the mode of action of several pairs of NLR that have been described to date. And indeed, this hypothesis is attractive and deserved to be tested. Authors tested all the other genes in the Rpp6907 locus (R1 to R6) by coexpressing each of them with Rpp6907-7. Here, it would have been very informative to show these data (NLR-triggered cell death response in *N. benthamiana* and immunoblot assays to verify the proper expression of the different epitope-tagged NLR candidates)

Response to the questions one by one: Thank you for your summary. We are extremely grateful to reviewer for pointing out the problem. The data all of the other genes in the Rpp6907 locus (R1 to R6) by coexpressing each of them with Rpp6907-7 (NLR-triggered cell death response in *N. benthamiana*) have been shown in Fig S8. The utilization of fusion GFP was employed for the purpose of detecting the typical expression of the target protein.

Q4. Interestingly, authors found that co-expression of Rpp6907-7 with Rpp6907-4 led to the inhibition of Rpp6907-triggered cell death, suggesting that Rpp6907-4 might function in pair with Rpp6907-7 to maintain the receptor complex in an inactive state. As mentioned above, I have the same comments to make to the authors regarding their interpretation of the presented data. In Fig4A, please show by immunoblots that

both NLR proteins are properly expressed and migrates at the expected size. Same in Fig 4D and 4E. In Fig4F-4G, important negative controls and immunoblots are missing (see my previous comments for Fig3)

Response to the questions one by one: Thank you for your suggestion. We have supplemented negative controls and immunoblots in Fig4F-4G. We have supplemented immunoblots in Fig4A.

Q5. Overall, I think that involvement of Rpp6907-4 in suppression of Rpp6907-7-mediated resistance should be supported with some other (genetic?) data. According to their model, in soybean, does Rpp6907-mediated resistance require Rpp6907-4? This could/should be investigated through RNAi to specifically silence Rpp6907-4 expression and check if it does compromise or not Rpp6907-7-mediated resistance

Response to the questions one by one: Thank you for your suggestion. To determine whether Rpp6907-mediated resistance require Rpp6907-4, we generated virus-induced gene silencing (VIGS) constructs based on recombinant tobacco rattle virus (TRV) to target Rpp6907-4 expression in SX6907. Our results showed that Rpp6907-7-mediated resistance was no reduction in TRV:Rpp6907-4-treated SX6907(Fig S9).

Reviewer #3

Q1.For the most part the data is clear and appropriate controls are included. However, this is not the case for the studies involving expression in *N. benthamiana*, where Western blot data is needed to confirm the expression of truncated proteins and mutant proteins that do not elicit cell death. Also, virtually no description and no sequence is provided for Rpp6907-4. Importantly, the authors should also confirm that Rpp6907-4 from the susceptible lines used for Rpp6907-7 expression (Tianlong1 and 06KG) can also suppress the cell death induced by Rpp6907-7.

Response to the questions one by one: Thank you for your suggestion. As suggested by reviewer, we have supplemented Rpp6907-4 description and sequence in Fig.4 and Table S5. We have cloned Rpp6907-4 like gene(W12) from the susceptible lines Tianlong1. We found that the co-expression of Rpp6907-7 and Rpp6907-4-like gene(W12) from Tianlong1 also attenuated or completely abolished Rpp6907-triggered HR (Fig. S11).

Q2. Line 29-The assertion that only SX6907 is the only soybean cultivar that has an immune response (IR) to ASR needs to be clarified or qualified. Rpp1 from PI 200492 is known to confer the IR to select isolates of *P. pachyrhizi*. In the reference cited (#3) only a single isolate was used to confirm the IR in SX6907.

Response to the questions one by one: We are extremely grateful to reviewer for pointing out this problem. We have corrected the inaccurate statement.

Q3. Line 35-the description of the roll of Rpp6907-4 as an inactive signaling executer is not clear to me, and the mention of non-canonical domains is equally puzzling.

Response to the questions one by one: Thank you for your suggestion. As suggested by reviewer, the statement in line 35 has been revised(Genetic analysis showed that

only Rpp6907-7 confers resistance, while Rpp6907-4 regulates Rpp6907-7 signaling activity by acting as a repressor in the absence of recognized effectors).

Q4. Line 59-are there food safety concerns associated with soybean rust?

Response to the questions one by one: Thank you for your question. What I want to say is that crop production and pesticide residues caused by plant diseases might cause food security and people's health. We regret for an unclear formulation in the original manuscript. We have removed this confusing part of the sentence, and hope that the current formulation is more clear.

Q5. Line 60-it would be better to first discuss the resistance genes from soybean (Rpp1-7) before introducing ccRpp1.

Response to the questions one by one: Thank you for your suggestion. As suggested by reviewer, the manuscript has been revised in Line 60.

Q6. Line 75-The claim of broad-spectrum resistance conferred by SX6907 needs to be supported-the reference cited used only one isolate.

Response to the questions one by one: We agree with the reviewer that this was an incorrect formulation. We have made corrections.

Q7. Line 89-The description of the location of R1- R6 and R7 could be simplified by indicating the genes flanking R1-R6 and R7

Response to the questions one by one: Thank you for your suggestion. As suggested by reviewer, the manuscript has been revised.

Q8. Line 98-Referring to R1- R6 and R7 as “intact” is confusing. Perhaps just drop “intact” as the length of the predicted proteins is stated. Likewise, provide the predate length of R4 instead of just saying it is truncated

Response to the questions one by one: Thank you for your suggestion. As suggested by reviewer, the manuscript has been revised.

Q9. Line 117 change heterozygous to hemizygous (correct in figure)

Response to the questions one by one: Thank you for your suggestion. As suggested by reviewer, the manuscript has been revised.

Q10. Line 134 provide more detail about the *P. pachyrhizi* populations used. I did not see this in the materials and methods.

Response to the questions one by one: Thank you for your suggestion. As suggested by reviewer, *P. pachyrhizi* populations information is supplemented in the material method.

Q11. Line 192... contains a single 2,571 bp exon and no introns

Response to the questions one by one: Thank you for your suggestion. As suggested by reviewer, the manuscript has been revised.

Q12. Line 247 It's not clear what is meant by “in soybean cells that are absent from *N. benthamiana*”

Response to the questions one by one: Thank you for your suggestion. As suggested by reviewer, the manuscript has been revised.

Q13. Line 275-does this imply that Rpp6907-4 is involved in the recognition event?

Response to the questions one by one: Thank you for your question. This is one possible mechanism but it is mere conjecture it has not been established by experiment.

Thank you for your careful review. We really appreciate your efforts in reviewing our manuscript. We wish good health to you and your family. Your careful review has helped to make our study clearer and more comprehensive.

Reviewers' Comments:

Reviewer #1:

Remarks to the Author:

It is still puzzling to me the reduction of expression level even in non-inoculated conditions, and the subcellular co-localization should also be studied more in detail, and that of Rpp6907-7 and Rpp6907-4 must be revisited in the full-length scope. Nevertheless, the genetic data suggests that Rpp6907-7 and Rpp6907-4 are paired NLRs against *P. pachyrizi*, and some limited evidences on molecular mechanism of repression is shown. I would like to thank the authors for making their best efforts to respond to my comments. The Rpp6907-7 and Rpp6907-4 model system is unique and I look forward to further discoveries regarding the mechanism of this system, such as identifying the *Pp* effector.

Reviewer #2:

Remarks to the Author:

I appreciate the authors' efforts to respond to my criticisms and questions. However, for almost all the controls that have been added in this new version (and whose absence I deplored in the previous version of the manuscript), the way in which they have been carried out or formatted is not satisfactory.

Below, I detail the various points that need to be rectified in order to make the authors' conclusions more convincing.

Fig 3E : immunoblots showing the proper accumulation of the different proteins (ie tagged with GFP) are still missing. The confocal image of the GFP fluorescence signal does not necessarily reflect the correct expression of GFP-fused proteins. Detection of the GFP signal could be due to free GFP resulting from a cleavage of the protein of interest in planta (instability, self cleavage...). Correct expression of full-length GFP-fused proteins in planta should be checked by immunoblot.

Fig 3F. An important negative control is still missing. Please include CC-AD co-expressed with an unrelated bait (BD) to check that CC-AD is not responsible for autoactivation of yeast reporter gene(s). Concerning the blots, they should not be performed like that. For ALL the combinations tested, authors should check the expression of both the prey and the bait proteins. In other words, for a given combination (prey+bait), the corresponding yeast protein extract should be probed with anti-HA and anti-myc antibodies. Control loading should be included (ponceau staining showing similar amount of total protein loaded in each lane). Please indicate somewhere (in the legend) the expected size of the different bait and prey proteins tested.

Fig 3G: From what I understand, the YFPN and YFPC fusion proteins are detected with an anti-HA and anti-Flag antibodies, respectively. Correct? Please add this information in the legend. For which reason the EV-YFPC control gives a signal similar in size to that of Rpp6907-7CC-YFPC? Does it correspond to YFPC alone? If so, the corresponding band should migrate to a lower size on the blot performed with anti-Flag.

Fig4A: immunoblot showing the expression of the different proteins fused with GFP or mCherry should be presented. I refer to the *N. benthamiana* samples corresponding the different combinations of *Agrobacterium* infiltrated. All the corresponding protein extracts should be probed with both anti-GFP AND anti-mCherry antibodies (with a loading control).

Fig 4G: Authors did not check whether Rpp6907-4-LRR-AD was autoactive in yeast. They should co-express this prey protein with an unrelated bait protein (BD). Expression of Rpp6907-CC-BD should be checked by immunoblot for ALL the combinations tested. Please also include a loading control.

Fig 4F: Immunoblots presented are not satisfying. All protein combinations tested should be probed with both anti-HA and anti-Flag antibodies. Expression of Rpp6907-4LRR-YFPC is not shown. Please include a loading control.

Fig S4 and S5. Immunoblots showing the accumulation of the different GFP fusion proteins is missing. Same comment as for Fig3E. All these proteins transiently expressed in *N. benthamiana* and tagged

with GFP should not be difficult to detect by immunoblot.

Fig S7. A and B letters to distinguish A and B panels are missing. S7A: it seems that part of the picture is identical to Fig3F. Please remove it from Fig S7A. Same for the immunoblot? Immunoblot data are partial (see my comments related to Fig 3F); S7B: same comment as for Fig 4F. Please include a loading control.

Fig S8: Immunoblots are missing. Same comment as for Fig 3E.

Fig S11. Please include an immunoblot showing the expression of the different proteins considered (anti-GFP/anti-mCherry, and with a loading control). The title of the figure is inconsistent with the numbers indicated in red (in brackets). Or did I miss something?

Reviewer #3:

Remarks to the Author:

I commend the authors for their willingness to incorporate the suggestions of the reviewers. The revised version of the manuscript is much improved. However, there are still a few items that require some clarification and/or editing. I will restrict my comments to the edits made to my previous review (Reviewer 3).

First, although it was not pointed out by any of the reviewers in the first review, the title needs to be changed. The word "resistance" is missing. Also, I don't think the word "novel" is necessary. (An atypical NLR pair confers Asian soybean rust resistance in soybean).

Q1: OK, but Table S5 is incorrectly labeled as S6 in the supplemental file.

Q2: OK, this is an important qualification to make.

Q3: OK

Q4: OK

Q5: OK, but verb tenses are incorrect.

Q6: Just like in the opening paragraph, the statement about complete resistance against *P. pachyrhizi* (lines 75 – 76) needs to be qualified (*P. pachyrhizi* isolate SS4).

Q7: As far as I can tell, this line has not been corrected. The description of the position of R1 – R7 is still not clear. Is the gene (the aspartyl protease) between R1 – R6 cluster and R7 a duplication or partial duplication of the gene 5' of R1? It is listed as 283100 in both places.

Q8: I appreciate the change, but the new sentence about R4 needs to be corrected: (R4 is truncated and predicted to encode an NLR containing 425 amino acids with an incomplete CC domain.

Q9: OK

Q10: Line 134 refers to ASR populations, but in the Materials & Methods under pathogen assays they are listed as ARS isolates from US and BR (Brazil). A distinction should be made as to whether these are populations or (purified) isolates. Also, unless these isolates have previously been reported and can be referenced, some additional information should be provided. Which isolates were collected in the US, which from Brazil, date of collection, etc.

Q11: OK

Q12: OK, much better.

Q13: This section regarding Rpp6907-4 and potential integrated decoys is still not clear to me. Although I realize it just conjecture, it is not clear whether Rpp6907-4 in other soybean lines have domains that may serve as integrated decoys.

Dear reviewers,

Firstly, we would like to thank you for your constructive comments concerning our article (Manuscript No.: NCOMMS-23-37156A). These comments are all valuable and helpful for improving our article. I really appreciate all your comments and suggestions! Please find my itemized responses in below and my revisions/corrections in the re-submitted files. All changes in the manuscript text file with colour highlighting.

Thanks again!

COMMENTS TO THE AUTHOR:

Reviewer #1:

Q1. It is still puzzling to me the reduction of expression level even in non-inoculated conditions, and the subcellular co-localization should also be studied more in detail, and that of Rpp6907-7 and Rpp6907-4 must be revisited in the full-length scope. Nevertheless, the genetic data suggests that Rpp6907-7 and Rpp6907-4 are paired NLRs against *P. pachyrhizi*, and some limited evidences on molecular mechanism of repression is shown. I would like to thank the authors for making their best efforts to respond to my comments. The Rpp6907-7 and Rpp6907-4 model system is unique and I look forward to further discoveries regarding the mechanism of this system, such as identifying the Pp effector.

Response to the questions one by one: Thank you for your recognition of our work. We appreciate the time and effort you put into reviewing our manuscript and providing valuable advice for improvement. We are currently studying the Rpp6907-7 and Rpp6907-4 model system to understand its underlying mechanism. Additionally, we have conducted whole genome sequencing and secreted proteome analysis to identify Pp effector proteins. We have predicted a group of candidate effectors and are actively working on this research. We are excited about the potential for positive results in the future and eagerly await your further guidance. Thanks again!

We found that compared to non-inoculated, after inoculated, there was no significant change in the expression level of Rpp6907-7, and the expression level of Rpp6907-4 increased after inoculated. Our results demonstrate that only Rpp6907-4 is induced by SS4. In addition, since we used isolated leaves for inoculation, there may be differences in gene expression at different times. As the time of leaf isolation increases, the expression levels of some genes may change, but no significant differences were found in our results. About the subcellular co-localization of Rpp6907-7 and Rpp6907-4, We used full-length Rpp6907-7 with GFP- tag and full-length Rpp6907-4 with mcherry-tagged for subcellular localization, and we observed subcellular localization more carefully through plasmolysis. When Rpp6907-7 and Rpp6907-4 are co expressed with the plasma membrane marker, respectively, Rpp6907-7-GFP is completely co-localized with PM-mCherry on all sides as a single layer, while Rpp6907-4-GFP is surrounded by PM-mCherry fluorescence signals and mainly expressed in the nucleus and cytoplasm. However, when Rpp6907-7-GFP is co expressed with Rpp6907-4-mCherry, the signal of Rpp6907-4-mCherry changes from the cytoplasm to the plasma membrane, completely overlaps with Rpp6907-7-GFP around the cell). So we hypothesize that they play a role in the plasma membrane (See below).

Reviewer #2

Q1. Fig 3E : immunoblots showing the proper accumulation of the different proteins (ie tagged with GFP) are still missing. The confocal image of the GFP fluorescence signal does not necessarily reflect the correct expression of GFP-fused proteins. Detection of the GFP signal could be due to free GFP resulting from a cleavage of the protein of interest in planta (instability, self cleavage...). Correct expression of full-length GFP-fused proteins in planta should be checked by immunoblot.

Response to the questions one by one: Thanks a lot for the very helpful and instructive comments. According to your suggestions, we have checked correct expression of full-length GFP-fused proteins in planta by immunoblot. The results were shown in Fig.3E.

Q2. Fig 3F. An important negative control is still missing. Please include CC-AD co-expressed with an unrelated bait (BD) to check that CC-AD is not responsible for autoactivation of yeast reporter gene(s).

Concerning the blots, they should not be performed like that. For all the combinations tested, authors should check the expression of both the prey and the bait proteins. In other words, for a given combination (prey+bait), the corresponding yeast protein extract should be probed with anti-HA and anti-myc antibodies. Control loading should be included (ponceau staining showing similar amount of total protein loaded in each lane). Please indicate somewhere (in the legend) the expected size of the different bait and prey proteins tested.

Response to the questions one by one: Thank you for your valuable review and suggestions about our manuscript. We have included CC-AD co-expressed with an unrelated bait (SS12-BD) to check that CC-AD is not responsible for autoactivation of yeast reporter gene(s). According to your suggestion, we performed an immunoblot for each yeast two-hybrid combination, and have included control loading. The expected size of the different bait and prey proteins tested have been indicated in the figure caption. The results were shown in Fig.3F.

Q3. Fig 3G: From what I understand, the YFPN and YFPC fusion proteins are detected with an anti-HA and anti-Flag antibodies, respectively. Correct? Please add this information in the legend.

For which reason the EV-YFPC control gives a signal similar in size to that of Rpp6907-7CC-YFPC? Does it correspond to YFPC alone? If so, the corresponding band should migrate to a lower size on the blot performed with anti-Flag.

Response to the questions one by one: We are extremely grateful to reviewer for pointing out this problem. I have corrected in Fig 3G.

Q4. Fig4A: immunoblot showing the expression of the different proteins fused with GFP or mCherry should be presented. I refer to the *N. benthamiana* samples corresponding the different combinations of *Agrobacterium* infiltrated. All the corresponding protein extracts should be probed with both anti-GFP AND anti-mCherry antibodies (with a loading control)

Response to the questions one by one: Thank you for your suggestions. we performed an immunoblot for each combination, and have included control loading. The expected size of the different proteins tested have been indicated in the figure caption. The results were shown in Fig.4A.

Q5. Fig 4F: Authors did not check whether Rpp6907-4-LRR-AD was autoactive in yeast. They should co-express this prey protein with an unrelated bait protein (BD). Expression of Rpp6907-CC-BD should be checked by immunoblot for ALL the combinations tested. Please also include a loading control.

Response to the questions one by one: Thank you for your suggestions. We have included Rpp6907-4-LRR-AD co-expressed with an unrelated bait (SS12-BD) to check that Rpp6907-4-LRR-AD is not responsible for autoactivation in yeast . We performed an immunoblot for each yeast two-hybrid combination, and have included control loading. The expected size of the different bait and prey proteins tested have been indicated in the figure caption. The results were shown in Fig.4F.

Q6. Fig 4G: Immunoblots presented are not satisfying. All protein combinations tested should be probed with both anti-HA and anti-Flag antibodies. Expression of Rpp6907-4LRR-YFPC is not shown.

Response to the questions one by one: Thank you for your suggestions. we performed an immunoblot for each combination, and have included control loading. The expected size of the different proteins tested have been indicated in the figure caption. The results were shown in Fig.4G.

Q7. Fig S4 and S5. Immunoblots showing the accumulation of the different GFP fusion proteins is missing. Same comment as for Fig3E. All these proteins transiently expressed in *N. benthamiana* and tagged with GFP should not be difficult to detect by immunoblot.

Response to the questions one by one: Thank you for your suggestions. we performed an immunoblot for each fusion protein, and have included control loading. The expected size of the different proteins tested have been indicated in the figure caption. The results were shown in Fig S4 and S5.

Q8. Fig S7. A and B letters to distinguish A and B panels are missing. S7A: it seems that part of the picture is identical to Fig3F. Please remove it from Fig S7A. Same for the immunoblot? Immunoblot data are partial (see my comments related to Fig 3F); S7B: same comment as for Fig 4F. Please include a loading control.

Response to the questions one by one: Thank you for your suggestions. We have revised Fig S7

accordingly.

Q9. Fig S8: Immunoblots are missing. Same comment as for Fig 3E.

Response to the questions one by one: Thank you for your suggestions. We have revised Fig S8 accordingly.

Q10. Fig S11. Please include an immunoblot showing the expression of the different proteins considered (anti-GFP/anti-mCherry, and with a loading control). The title of the figure is inconsistent with the numbers indicated in red (in brackets). Or did I miss something?

Response to the questions one by one: Thank you for your suggestions. We have revised Fig S8 accordingly.

Finally, we would like to thank you again for your profound help in this work and promoting the scientific significance of this manuscript. In summary we feel the comments/suggestions/corrections to be very insightful and very helpful. Indeed, we feel the paper becomes much more solid and stronger after revision by considering your comments.

Reviewer #3

Q1. First, although it was not pointed out by any of the reviewers in the first review, the title needs to be changed. The word “resistance” is missing. Also, I don’t think the word “novel” is necessary. (An atypical NLR pair confers Asian soybean rust resistance in soybean).

Response to the questions one by one: Thank you for your suggestions. We have revised title accordingly.

Q2. Table S5 is incorrectly labeled as S6 in the supplemental file.

Response to the questions one by one: Sorry for the mistake. We have corrected the labeling.

Q3. OK, but verb tenses are incorrect.

Response to the questions one by one: Sorry for the mistake. We have corrected the text.

Q4. Just like in the opening paragraph, the statement about complete resistance against *P. pachyrhizi* (lines 75-76) needs to be qualified (*P. pachyrhizi* isolate SS4).

Response to the questions one by one: Thank you for your suggestions. We have revised the text accordingly.

Q5. As far as I can tell, this line has not been corrected. The description of the position of R1- R7 is still not clear. Is the gene (the aspartyl protease) between R1-R6 cluster and R7 a duplication or partial duplication of the gene 5’ of R1? It is listed as 283100 in both places.

Response to the questions one by one: Sorry for the mistake. We have corrected the text accordingly.

Q6. I appreciate the change, but the new sentence about R4 needs to be corrected: (R4 is truncated and predicted to encode an NLR containing 425 amino acids with an incomplete CC domain).

Response to the questions one by one: Thank you for your suggestions. We have revised the text accordingly.

Q7. Line 134 refers to ASR populations, but in the Materials & Methods under pathogen assays they are listed as ARS isolates from US and BR (Brazil). A distinction should be made as to whether these are populations or (purified) isolates. Also, unless these isolates have previously been reported and can be referenced, some additional information should be provided. Which isolates were collected in the US, which from Brazil, date of collection, etc.

Response to the questions one by one: Sorry for the mistake. We have corrected the text. We

used ARS populations from US and BR (Brazil).

Q8. This section regarding Rpp6907-4 and potential integrated decoys is still not clear to me. Although I realize it just conjecture, it is not clear whether Rpp6907-4 in other soybean lines have domains that may serve as integrated decoys.

Response to the questions one by one: Thank you for your suggestions. We analyzed the domains of Rpp6907-4-like in other soybean lines (28 soybean accessions in Fig.2), and did not find any typical domains other than CC, NB-ARC and LRR. But we found that they had an AIF-C domain, further research is under way.

Once again, we sincerely appreciate the insightful comments from you that have been very helpful in guiding us along our revision.

Reviewers' Comments:

Reviewer #1:

Remarks to the Author:

Compared to the first draft, there has been improvement. I have some more minor comments in terms of texts for edits.

1. The question concerning me was not that there is no difference in expression between the inoculated and non-inoculated, but that there is increased expression at 1h post treatment. This perhaps could be a DAMP effect.

Line 73

an urgent need -> there is an urgent need

Line 93

Summary -> In summary,

Line 103

For the incomplete CC domain, is the truncation at the N-terminal domain? It also seems to have shorter LRR domain, how many LRR repeats were predicted?

Line 267

The term 'regulates' is too broad. I would suggest to use Rpp6907-4 represses Rpp6907-7 autoactivation in *Nicotiana benthamiana*.

Line 278

In order to find the helper of Rpp6907-7

-> this should be changed. The nomenclature agreed upon is that RRS1 is the sensor NLR, whereas RPS4 will be the helper NLR.

Line 305

I cannot understand the reason for S10. The distances between these sensor NLRs are too far apart to have any useful meaning. Please delete. In addition, there are additional paired NLRs with truncated forms of sensor NLRs, without integrated domains.

Line 518

Please use up-to-date reference for BiFC vectors.

Line 566

Please provide catalogue number and if available, the LOT number for the antibodies.

Reviewer #2:

Remarks to the Author:

My concerns have been addressed. Thanks. I am satisfied with the current version of the revised manuscript, and have no further comments.

Reviewer #3:

Remarks to the Author:

I am satisfied with the responses to my suggestions and questions. However, I still find that the statement regarding NUMEROUS potential integrated decoys in Rpp6907-4 homologs lacks support from the data presented (lines 306-309).

"Q8. This section regarding Rpp6907-4 and potential integrated decoys is still not clear to me.

Although I realize it just conjecture, it is not clear whether Rpp6907-4 in other soybean lines have domains that may serve as integrated decoys.

Response to the questions one by one: Thank you for your suggestions. We analyzed the domains of Rpp6907-4-like in other soybean lines(28 soybean accessions in Fig.2), and did not find any typical

domains other than CC, NB-ARC and LRR. But we found that they had an AIF-C domain, further research is under way."

If I understand correctly, Rpp6907-4-like proteins in other soybean lines contain AIF-C domains, but no other typical domains. First, has the presence of AIF-C domains in Rpp6907-4-like proteins been added to the manuscript? And even so, how does the presence of just AIF-C domains support the conjecture?

Dear Reviewers,

Firstly, we would like to thank you for your kind letter and for the reviewers' constructive comments concerning our article (Manuscript No.:NCOMMS-23-37156A). These comments are all valuable and helpful for improving our article. I really appreciate all your comments and suggestions! Please find my itemized responses in below and my revisions/corrections in the re-submitted files. All changes in the manuscript text file with colour highlighting.

Thanks again!

COMMENTS TO THE AUTHOR:

Reviewer #1:

Q1. Compared to the first draft, there has been improvement. I have some more minor comments in terms of texts for edits.

1. The question concerning me was not that there is no difference in expression between the inoculated and non-inoculated, but that there is increased expression at 1h post treatment. This perhaps could be a DAMP effect.

Response to the questions one by one: Thanks for your affirmation of the execution and results for our work. Thanks a lot for the very helpful and instructive comments. About the question "there is increased expression at 1h post treatment", we found that some genes in 'regulation of jasmonic acid mediated signaling pathway', 'regulation of defense response', 'response to wounding', 'regulation of response to stress' pathway were most significantly induced by RNA-seq data analysis, suggesting a potential DAMP effect. We will investigate this further in future research.

Gene Ontology term	Cluster frequency	Genome frequency of use	Corrected P-value
regulation of jasmonic acid mediated signaling pathway (view genes)	5 out of 180 genes, 2.8%	30 out of 28666 genes, 0.1%	0.00039
regulation of defense response (view genes)	6 out of 180 genes, 3.3%	66 out of 28666 genes, 0.2%	0.00126
response to wounding (view genes)	5 out of 180 genes, 2.8%	49 out of 28666 genes, 0.2%	0.00475
regulation of response to stress (view genes)	6 out of 180 genes, 3.3%	83 out of 28666 genes, 0.3%	0.00481
negative regulation of RNA biosynthetic process (view genes)	6 out of 180 genes, 3.3%	134 out of 28666 genes, 0.5%	0.07039
negative regulation of nucleic acid-templated transcription (view genes)	6 out of 180 genes, 3.3%	134 out of 28666 genes, 0.5%	0.07039
inositol biosynthetic process (view genes)	2 out of 180 genes, 1.1%	4 out of 28666 genes, 0.0%	0.07886
negative regulation of RNA metabolic process (view genes)	6 out of 180 genes, 3.3%	138 out of 28666 genes, 0.5%	0.08253
negative regulation of nucleobase-containing compound metabolic process (view genes)	6 out of 180 genes, 3.3%	153 out of 28666 genes, 0.5%	0.14345
toxin catabolic process (view genes)	4 out of 180 genes, 2.2%	59 out of 28666 genes, 0.2%	0.17665
secondary metabolite catabolic process (view genes)	4 out of 180 genes, 2.2%	59 out of 28666 genes, 0.2%	0.17665
toxin metabolic process (view genes)	4 out of 180 genes, 2.2%	60 out of 28666 genes, 0.2%	0.18835
polyol biosynthetic process (view genes)	2 out of 180 genes, 1.1%	6 out of 28666 genes, 0.0%	0.19552
copper ion transmembrane transport (view genes)	3 out of 180 genes, 1.7%	28 out of 28666 genes, 0.1%	0.24017
mRNA 5'-splice site recognition (view genes)	2 out of 180 genes, 1.1%	7 out of 28666 genes, 0.0%	0.27261
copper ion transport (view genes)	3 out of 180 genes, 1.7%	30 out of 28666 genes, 0.1%	0.29491
negative regulation of macromolecule biosynthetic process (view genes)	6 out of 180 genes, 3.3%	177 out of 28666 genes, 0.6%	0.30809
negative regulation of biosynthetic process (view genes)	6 out of 180 genes, 3.3%	179 out of 28666 genes, 0.6%	0.32652
negative regulation of cellular biosynthetic process (view genes)	6 out of 180 genes, 3.3%	179 out of 28666 genes, 0.6%	0.32652
negative regulation of nitrogen compound metabolic process (view genes)	6 out of 180 genes, 3.3%	190 out of 28666 genes, 0.7%	0.44355
secondary metabolic process (view genes)	10 out of 180 genes, 5.6%	511 out of 28666 genes, 1.8%	0.51713
L-serine biosynthetic process (view genes)	2 out of 180 genes, 1.1%	10 out of 28666 genes, 0.0%	0.57696
copper ion export (view genes)	2 out of 180 genes, 1.1%	12 out of 28666 genes, 0.0%	0.83924
inositol metabolic process (view genes)	2 out of 180 genes, 1.1%	15 out of 28666 genes, 0.1%	1
protein refolding (view genes)	2 out of 180 genes, 1.1%	15 out of 28666 genes, 0.1%	1
sesquiterpenoid metabolic process (view genes)	2 out of 180 genes, 1.1%	16 out of 28666 genes, 0.1%	1
mRNA splice site selection (view genes)	2 out of 180 genes, 1.1%	18 out of 28666 genes, 0.1%	1
mRNA cis splicing, via spliceosome (view genes)	2 out of 180 genes, 1.1%	18 out of 28666 genes, 0.1%	1
response to hypoxia (view genes)	2 out of 180 genes, 1.1%	19 out of 28666 genes, 0.1%	1
response to decreased oxygen levels (view genes)	2 out of 180 genes, 1.1%	19 out of 28666 genes, 0.1%	1
response to oxygen levels (view genes)	2 out of 180 genes, 1.1%	19 out of 28666 genes, 0.1%	1

Q2. Line 73

an urgent need -> there is an urgent need

Response to the questions one by one: Thank you for your suggestion. As suggested by reviewer, the manuscript has been revised.

Q3. Line 93

Summary -> In summary,

Response to the questions one by one: Thank you for your suggestion. As suggested by reviewer, the manuscript has been revised.

Q4. Line 103

For the incomplete CC domain, is the truncation at the N-terminal domain? It also seems to have shorter LRR domain, how many LRR repeats were predicted?

Response to the questions one by one: Thank you for your question. Rpp6907-4 CC domain, is the truncation at the N-terminal domain. And Rpp6907-4 have shorter LRR domain at the C-terminal. Two LRR repeats were predicted.

Q5. Line 267

The term 'regulates' is too broad. I would suggest to use Rpp6907-4 represses Rpp6907-7 autoactivation in *Nicotiana benthamiana*.

Response to the questions one by one: Thank you for your suggestion. As suggested by reviewer, the manuscript has been revised.

Q6. Line 278

In order to find the helper of Rpp6907-7

-> this should be changed. The nomenclature agreed upon is that RRS1 is the sensor NLR, whereas RPS4 will be the helper NLR.

Response to the questions one by one: Thank you for your suggestion. As suggested by reviewer, the manuscript has been revised.

Q7. Line 305

I cannot understand the reason for S10. The distances between these sensor NLRs are too far apart to have any useful meaning. Please delete. In addition, there are additional paired NLRs with truncated forms of sensor NLRs, without integrated domains.

Response to the questions one by one: Thank you for your suggestion. As suggested by reviewer, the manuscript has been revised.

Q8. Line 518

Please use up-to-date reference for BiFC vectors.

Response to the questions one by one: Thank you for your suggestion. As suggested by reviewer, the manuscript has been revised.

Q9. Line 566

Please provide catalogue number and if available, the LOT number for the antibodies.

Response to the questions one by one: Thank you for your suggestion. As suggested by reviewer, the manuscript has been revised.

Reviewer #2

Q1. My concerns have been addressed. Thanks. I am satisfied with the current version of the revised manuscript, and have no further comments.

Response to the questions one by one: Thanks for your affirmation of the execution and results for our work.

Reviewer #3

Q1. I am satisfied with the responses to my suggestions and questions. However, I still find that

the statement regarding NUMEROUS potential integrated decoys in Rpp6907-4 homologs lacks support from the data presented (lines 306-309).

"Q8. This section regarding Rpp6907-4 and potential integrated decoys is still not clear to me. Although I realize it just conjecture, it is not clear whether Rpp6907-4 in other soybean lines have domains that may serve as integrated decoys.

Response to the questions one by one: Thank you for your suggestions. We analyzed the domains of Rpp6907-4-like in other soybean lines(28 soybean accessions in Fig.2), and did not find any typical domains other than CC, NB-ARC and LRR. But we found that they had an AIF-C domain, further research is under way."

If I understand correctly, Rpp6907-4-like proteins in other soybean lines contain AIF-C domains, but no other typical domains. First, has the presence of AIF-C domains in Rpp6907-4-like proteins been added to the manuscript? And even so, how does the presence of just AIF-C domains support the conjecture?.

Response to the questions one by one: Thanks a lot for the very helpful and instructive comments. Because the statement regarding numerous potential integrated decoys in Rpp6907-4 homologs lacks support from the data presented (lines 306-309). This caused us to reconsider our speculation about the potential integrated decoys in Rpp6907-4, and we decided to remove the related text from the discussion. We are currently exploring this in more detail and will address this in future work.

Reviewers' Comments:

Reviewer #1:

Remarks to the Author:

My questions have been addressed. I thank the authors for their response.

Reviewer #3:

Remarks to the Author:

My remaining concerns have been addressed. I have no further comments, questions, or suggestions.

Dear Reviewers,

We are pleasure that our replies addressed the reviewer's concerns and thank reviewer for the affirmation. We also appreciate the kind and constructive comments from reviewer for improving our work.

Thanks again!